# Mustard Meal Extract as an Alternative to Zinc Oxide for Protecting the Intestinal Barrier Against *E. coli*-Lipopolysaccharide Damage

**DOI:** 10.3390/ijms26010273

**Published:** 2024-12-31

**Authors:** Ionelia Taranu, Cristina Valeria Bulgaru (Procudin), Gina Cecilia Pistol, Mihai Alexandru Gras, Ana-Maria Ciupitu, Iulian Alexandru Grosu, Mihaela Vlassa, Miuta Filip, Daniela Eliza Marin

**Affiliations:** 1Laboratory of Animal Biology, National Institute for Research and Development for Biology and Animal Nutrition, Calea Bucuresti No. 1, Balotesti, 077015 Ilfov, Romania; 2Raluca Ripan Institute for Research in Chemistry, Babeș-Bolyai University, 30 Fântânele Street, 400294 Cluj-Napoca, Romania

**Keywords:** mustard meal, endotoxin lipopolysaccharide, intestine epithelium, Caco-2/HT29-MTX cellular model, antioxidants, anti-inflammation, antibacterial

## Abstract

The present study aimed to investigate the ability of an aqueous extract derived from mustard seed meal to counteract the effects of *E. coli* endotoxin lipopolysaccharide (LPS) on the intestinal epithelium. Caco-2 cells were cultured together with HT29-MTX and used as a cellular model to analyze critical intestinal parameters, such as renewal, integrity, innate immunity, and signaling pathway. Byproducts of mustard seed oil extraction are rich in soluble polysaccharides, proteins, allyl isothiocyanates, and phenolic acids, which are known as powerful antioxidants with antimicrobial and antifungal properties. Cells were seeded at a ratio of nine (Caco-2) to one (HT29-MXT) and treated for 2 h with mustard meal extract (ME, dilution 1/50) and zinc oxide (ZnO, 50 μM) after reaching 80–100% confluence. Then, they were challenged with 5 μg/mL *E. coli*-LPS and incubated for another 4 h. The results show that LPS did not alter the cell viability but decreased proliferation compared to the control, ME and ZnO treatments. LPS altered the cell membrane integrity and monolayer permeability by decreasing the transepithelial electrical resistance and tight-junction protein expression. In addition, LPS increased the activity of LDH and the expression of Toll-like receptors. The mechanisms by which LPS induces these disturbances involves the overexpression of PKC, p38 MAPK, and NF-κB signaling molecules. The pretreatment with mustard meal and ZnO succeeded in counteracting the impairment of epithelial renewal, the damage of the membrane integrity and permeability as well as in restoring the gene expression of tight-junction proteins.

## 1. Introduction

The colon is a key part of the digestive system because of its main function of absorbing water and salts from solid food waste. It is also the site of bacterial fermentation of unabsorbed substances (e.g., dietary fiber and other fermentable substrates); therefore, colonic tissue is very sensitive to the quantity and quality of dietary intake [1,2]. This is especially important in piglets during the weaning period, as their digestive system is still immature and requires high-quality feedstuffs [3]. If the feed during this period is poor it can lead to gut disturbances (e.g., inflammation, improper fermentations) which mainly affects the colon [4]. Specific to piglets during the weaning period is also their immature immune system, which can neither manage the various pathogens that contaminate their feed, such as enterotoxigenic *Escherichia coli*, *Salmonella Thyphimurium*, and *Rotavirus*, nor their LPS endotoxins which also cause gut inflammation and intestinal barrier damage [5]. Lipopolysaccharides (LPSs) are characteristic components of the cell wall of Gram-negative bacteria. Chemically, LPSs are glycolipids composed of carbohydrates linked to fatty acids [6]. A typical LPS molecule consists of the following three parts: lipid A (which confers toxicity), an outer core polysaccharide (O-antigen), and an inner core oligosaccharide, all covalently linked. LPSs contribute to the integrity of the external membrane, protecting the bacterial cell against the action of bile salts and lipophilic antibiotics. It has been shown that LPSs are toxins that stimulate host cells, triggering inflammation by binding the CD14/TLR4/MD2 complex. This stimulation results in the release of pro-inflammatory cytokines, such as TNFα (tumor necrosis factor), IL-1 (interleukin 1), and IL-6 (interleukin 6), nitric oxide, and eicosanoids [7]. The mechanism involves the NF-kB pathway [8].

Inflammation and other disorders caused by LPSs have become critical issues in swine production, since the use of in-feed antibiotics and pharmacological zinc oxide (ZnO) as prophylactic antimicrobials and growth promoters was banned by the European Commission (2006 and 2023) because of increased bacterial resistance, with consequences for human health and the environmental [3,9]. For this reason, investigation of alternative, sustainable sources of bioactives, including probiotics, parabiotics, and postbiotics, as well as active phytocompounds with health-promoting biological properties, has become both necessary and of interest over the last decade [10]. Dietary supplementation with a probiotic, paraprobiotic, and a hydrolyzed yeast mixture had effects comparable to ZnO on growth performance and nutrient digestibility [11]. Along the same line, a combination of yeast beta-glucan [10] and milk hydrolysate (milk-derived sodium caseinate) proved to be a suitable alternative to ZnO with positive effects on growth performance, fecal consistency, microbiota, and inflammation. A previous study by Guilloteau [12] reported that Na-butyrate was able to replace the antibiotic flavomycin by enhancing the growth rate, feed conversion, gastrointestinal tract digestive capacities, and production of digestive enzymes. The mechanism by which this occurs is likely the implication of insulin-like growth factor-1 [12].

Other studies have also demonstrated the effectiveness of several plant-derived phytocompounds (fruits, vegetables, cereals, etc.) or their byproducts, which contain significantly high concentrations of bioactive compounds that can reduce inflammation and maintain intestinal health and epithelium integrity, renewal, and function. Taranu et al. [13] tested in vitro the capacity of an extract derived from fermented rapeseed meal to replace ZnO, with the results indicating a reduction in the production of ROS, as well as the inhibition of apoptosis and pro-inflammatory cytokines, in epithelial intestinal cells [13]. Other studies have also demonstrated the effectiveness of phytocompounds as replacements for both ZnO and antibiotics. Aqueous and solvent extracts of *Pulicaria crispa* showed antimicrobial activity against Gram-positive and Gram-negative pathogens, with the ethyl acetate extract being the most effective [14].

Mustard seed byproducts of mustard oil extraction are rich in thousands of bioactive compounds, among which allyl isothiocyanates stand out because of their antimicrobial capacity [15]. Indeed, there are many studies that have demonstrated the strong antibacterial effect of mustard compounds (allyl isothiocyanates and essential oil) encapsulated in different materials against Gram-negative (*E. coli*, *B. subtilis*, *S. sonnei*, *S. lignieres*, *P. aeruginosa*, and *P. fluorescens*) or Gram-positive (*L. monocitogenes*, *S. aureus*, *M. luteus*, and *S. epidermidis*) bacteria [15,16,17]. In addition, phenolic acids derived from mustard like tannins, gallic, caffeic, cinnamic, benzoic acids were reported to have not only antimicrobial but also antioxidant, antifungal [18] as well as antimycotoxic activity [19,20].

However, the effect of mustard byproducts and its compounds on the intestinal epithelium in vitro or in vivo are not known. This is the reason the investigation of bioactive agents from less characterized sources represents an important subject for the fields of animal nutrition and health research.

In this study, a cellular model was used which mimics the intestinal physiological conditions consisting in the coculturing of two types of epithelial cells, colonocytes Caco-2 and mucous-secreting HT29-MTX cells. The aim was to investigate the potential efficacy of an aqueous extract derived from mustard seed meal in counteracting the deleterious effects produced by a short exposure to *E. coli* endotoxin lipopolysaccharide on the renewal, integrity, pathogen recognition markers (toll-like receptors) and related signaling pathway of the colonic epithelium.

## 2. Results

### 2.1. Total Polyphenolic Content and Antioxidant Activity of Mustard Seed Byproduct

The total polyphenol content of the mustard meal was 10.93 mg GAE/g mustard meal sample, with an antioxidant activity of 2881.84 μmol TE/100g (ABTS test).

#### In Vitro Assessment of Caco-2/HT29-MXT Monolayer Renewal

Epithelial renewal, a key characteristic of intestinal epithelia, was assessed through cell viability, proliferation, cycle progression, and apoptosis.

### 2.2. Cell Viability

Cell viability was measured when cells reached 80% confluence. Caco2/HT29-MTX cocultured cells were pretreated with either 1/50 ME or ZnO solution (50 µM) for 2 h and then challenged with LPS for another 4 h to analyze the effect of the mustard extract bioactives and ZnO on the cell viability. The MTT results show that, over the 4 h, the LPS did not affect the cellular viability (100.00 ± 3.86%) compared to the control (100.00% ± 0.00). The pretreatments with both ME and ZnO alone maintained the cellular viability close to that of the control, with a very slight increase being observed for ZnO (106.30 ± 6.31%). In cell cultures pretreated with ME and ZnO for 2 h, the addition of LPS did not affect their viability (104.00 ± 7.69% for ME+LPS and 106.37 ± 5.93% for ZnO+LPS, Figure 1) compared to the control or between the treatment.

### 2.3. Cell Proliferation

The nuclear antigen Ki67 is a marker of cell proliferation, as it is expressed only in cells undergoing division. In this study, Ki67 was measured, using the Muse™ Ki67 Proliferation Kit, in cocultured Caco2/HT29-MTX cells that had reached 80% confluence, were pretreated with 1/50 ME and ZnO (50 µM) for 2 h, and then challenged with LPS for another 4 h. Unlike the cell viability, LPS decreased epithelial cell proliferation compared to the control and experimental treatments (*p* < 0.0001). In contrast, a significant increase (*p* < 0.0001) in proliferation was observed for cells treated with either the mustard meal extract or zinc oxide alone, as well as in cocultures pretreated with ME and ZnO and challenged with LPS compared to the control and LPS (Figure 2).

### 2.4. Cell Apoptosis

The results, as shown in Figure 3, suggest that LPS induced the highest significant percentage of apoptosis (both early and late apoptosis) and cellular death in a coculture treated with endotoxin for 4 h compared to the control. In contrast, the ME and ZnO treatments alone induced much lower percentages of early and late apoptosis, closer to those of the control. In addition, pretreatment with ME and ZnO for 2 h attenuated the apoptotic effects caused by LPS, suggesting that ME and ZnO have a protective effect against LPS cytotoxicity. Of interest, ZnO alone induced a higher percentage of total apoptotic cells compared to the control or ME alone. The percentage of dead cells was higher for the ME+LPS and ZnO+LPS treatments.

### 2.5. Cell Cycle

The results from Figure 4 show that the percentage of cells in the G0/G1 resting phase decreased for all treatments compared with that of the control, and the decrease was significant not only for cells exposed for 4 h to LPS (−17.95%, *p* < 0.003) but also for the cells treated with ZnO alone (−15.68%, *p* < 0.006), or ZnO+LPS (−19.67%, *p* < 0.001), indicating that these treatments might predispose the cells for the apoptotic process. Indeed, these results are correlated with those obtained for apoptosis, which clearly showed that LPS and ZnO increased the percentage of cells in the late-apoptosis phase (Figure 4). During the DNA synthesis phase (S), only LPS alone altered the distribution of the cells, but the decrease is not significant. In contrast, cells treated with ME, ZnO or ME+LPS registered a slight increase in the percentage in the S phase. Of interest, the cells pretreated with ZnO and challenge with LPS registered the highest percentage of cells within the phase G2/M.

#### In Vitro Assessment of Caco-2/HT29-MXT Monolayer Integrity

The protective effect of mustard meal extract and ZnO on Caco2-HT29-MXT monolayer integrity against *E. coli*-LPS was investigated through evaluation of transepithelial resistance (TEER), tight-junction proteins expression, fluorescein isothiocyanate (FITC)-conjugated dextran 4 (FD4) and LDH activity.

### 2.6. Trans-Epithelial Electrical Resistance (TEER)

The TEER measured after 21 days of Caco-2/HT29-MTX cells cultivation dramatically decreased after 6 h of LPS treatment, from 1127.2 Ω at the time = 0 h to 831.0 Ω at the time = 6 h meaning a significant decreasing from 100% to 73% compared to control. TEER value at the time = 24, 48 and 72 h remained low very closed to that at 6 h in cells treated with LPS (around 800 Ω, Figure 5). In contrast, the TEER of Caco-2/HT29-MTX monolayer treated with ME, ZnO alone was similar to that of control at all four time points (between 99.21% and 100.53%). The two treatments were able to counteract the degradation of TEER produced by LPS and no difference between ME+LPS and ZnO+LPS was observed (Figure 5).

### 2.7. Modulation of the Gene Expression of Tight-Junction Proteins

Caco2/HT29-MXT cells were treated with ME and ZnO for 2 h and then exposed to *E. coli*-LPS for 4 h. The data in the literature show that LPS causes epithelial barrier damage by disrupting tight-junction proteins (TJs) [21]. As expected, in this study, significant (*p* < 0.05) suppression (one hundred percent, 9/9 genes) of the gene expression of CLDN-1, -2, -4, -5, -14, -20, -23, OCCL, and ZO1 was registered in a coculture monolayer challenged with 5 µg LPS/mL when compared to the control’s monolayer (Table 1). The most downregulated were CLDN-1 (−30% decrease vs. the control; Table 1) and CLDN-2 (also −30% decrease vs. the control, *p* < 0.05). Treatments with both ME and ZnO alone had no effect on five of the nine TJs genes (CLDN-4, -5, -14, -20, and -23) compared to the control, and both counteracted the downregulation effect of LPS, restoring and upregulating the gene expression of tight junctions (Table 1). Furthermore, both ME and ZnO induced significantly large increases in the gene expression of CLDN-1 (1.7 fold change- fc), CLDN-2 (1.6 fold change), and CLDN-14 (1.8 fold change), with a pronounced induction of CLDN-1 (3.3 fc) under the action of the ZnO treatment (Table 1).

The results show that the expression of regulatory TJ genes was also drastically downregulated by LPS compared to the control (6/6). The most affected were the myosin light-chain kinase (MLCK) gene (0.2 fc, −80%), followed by the membrane-associated guanylate kinase (MAGI2) gene (0.4-fc decrease, −60%) and the tyrosine-protein phosphatase nonreceptor type 2 (PTPN2) gene (0.4-fc decrease, −60%). The ME and ZnO treatments maintained the expression of these genes close to that of the control. Compared with the control, the LPS and ME treatments, as well as ZnO, remarkably increased (3.9-fold) the expression of GNAI2 (guanine nucleotide-binding protein G(i), alpha-2 subunit), a gene involved in various transmembrane signaling systems as a modulator or transducer. The high upregulation of GNAI2 was also maintained with ZnO in combination with LPS (3.5-fc increase). However, both ME and ZnO restored the expression of regulatory TJ genes (4/6—ME and 5/6—ZnO; Table 1) compared to LPS.

To validate the gene profiles obtained by the qPCR array, the protein expression level of claudin-4 (CLDN4) was measured by immunofluorescence in all samples used for the qPCR array (Figure 6). The expression of the CLDN-4 protein showed a similar expression pattern as that obtained from the qPCR results (Figure 6).

### 2.8. Cell Monolayer FD4 Permeability After ME and ZnO Treatments and LPS Challenge

To test the monolayer permeability, Caco2/HT29-MXT cells were cultured for 3 days and then treated with ME and ZnO for 2 h and challenged with LPS for another 4 h. The results, as shown in Table 2, indicate no change in the transport of fluorescein isothiocyanate (FITC)-conjugated dextran 4 (FD4) in the control and individual ME and ZnO treatments, with FD4 being maintained in the apical compartment. In contrast, the concentration of FD4 increased 3.26-fold in the basal compartment of the culture treated for 4h with LPS, suggesting a significant disruption in the monolayer’s integrity, most likely through LPS-induced tight-junction protein damage. The ME and ZnO treatments alone or after the addition of LPS successfully maintained the epithelial barrier’s integrity, with a higher concentration of FD4 in the apical pole and a lower one in the basal pole (Table 2).

### 2.9. LDH Release Assay

The *E. coli*-LPS-induced membrane damage was also estimated by measuring the LDH activity. The results, as shown in Figure 7, indicate a markedly significant difference in the release of LDH between the control cells and the LPS-treated cells (2.96-fold increase for LPS). In contrast, in the cells pretreated with ME and ZnO for 2 h, LDH decreased significantly, particularly for ME (−42.37% vs. LPS). The mustard extract seems to be more effective than ZnO (−21.80% vs. LPS). Individually, these treatments had LDH values close to those of the control (Figure 7).

### 2.10. Innate Immune Modulation

Zinc oxide is known and used for its antimicrobial potential. Its effect, compared to those of mustard extract and LPS endotoxin, was evaluated in the Caco2/HT29-MXT monolayer by measuring the gene expression of TLRs, which are the main components of the innate immune response and responsible for the initial interaction with pathogens. In agreement with the working hypothesis, LPS significantly increased the gene expression of all investigated TLRs (TLR-1 to -10; Table 3), with the gene for TLR-4, the receptor that specifically recognizes LPS, being the most overexpressed (6.5-fc). In the cell culture exposed to the ME treatment alone, the gene expression of all TLRs was close to that of the control, whereas ZnO slightly but not significantly decreased the expression of three TLRs (Table 3). Both treatments were able to restore the expression of TLRs genes to the control level. LPS exerted a similar influence, increasing the gene expression of some mediator molecules associated with TLR pathways (Table 4), whereas the pretreatments with ME and ZnO reduced the upregulation of these genes induced by LPS (Table 4).

### 2.11. Mitochondrial Morphology and Membrane Potential

The results in Figure 8 show that the mitochondrial morphology of the cells challenged with LPS changed, exhibiting unclear, irregular shapes compared to the control cells, suggesting a decrease in the mitochondrial membrane potential. In contrast, in the cells treated with either ME or ZnO alone, ME+LPS, or ZnO+LPS, mitochondria appeared bright in color with an intact/visible membrane and normal morphology. As regard the mitochondrial potential, Table 5 shows that the highest percentage of depolarized cells, both live and dead, was observed after exposure to LPS compared to the control. In contrast, in the cocultured cells pretreated with ME and ZnO alone or in combination with LPS, the mitochondrial membrane potential significantly decreased (*p* < 0001; Table 5).

### 2.12. Signaling Pathway

One of the most important effects of LPS is inflammation. At the intestinal level, inflammation causes functional impairment of tight-junction proteins and pro-inflammatory cytokine synthesis. The mechanism that triggers inflammation is the stimulation of the NF-kB signaling pathway. Indeed, in this study, the gene expression of NF-kB was the most overexpressed by LPS (3.6 fc vs. control), and overexpression was also observed for TAK-1 (1.8 fc) and p-38 MAPK (2.28 fc) due to the influence of LPS compared to the control. As expected, LPS increased the genes coding for PKC-A (1.8 fc) and PKC-B (2.2 fc), two kinases implicated in the modulation of tight-junction functionality, an effect that could lead to the loss of their ability to form a tight epithelial barrier. The ability of ME and ZnO to counteract LPS was observed at the signaling pathway level, downregulating the LPS-induced gene expression of NF-kB, MAPK-p38, and other kinases to control levels (Table 6).

## 3. Discussion

Interventions in the nutrition of farm animals aimed at improving health and the immune defense response have significantly increased over the last decade after the ban on in-feed antibiotics as growth promoters and pharmaceutical ZnO utilization. Unfortunately, although aiding in the reduction of the rise in bacterial resistance to antibiotics, it can also lead to an increase in bacterial occurrences and, consequently, to an increase in the number of infections in farm animals, with harmful effects at the intestinal level, especially during the weaning period [22,23]. In response to this problem, many analytical studies have shown that various raw materials, such as agri-food byproducts, are rich sources of bioactive compounds with antimicrobial properties that can replace antibiotics successfully. Along this line, in the present study, we investigated the potential efficacy of an extract derived from mustard meal to counteract intestinal barrier dysfunctions produced by *E. coli*-LPS using a complex cellular model that involves two of the most important cell types in intestinal epithelia, Caco-2 (absorptive enterocytes) and HT29-MTX (mucus-producing cells). The model mimics the epithelium and intestinal barriers better than Caco-2 culture alone. The intestinal epithelium undergoes constant and rapid renewal, a process that contributes to the repair of mucosal injuries and ensures the integrity of the epithelium [24]. The renewal of the intestinal epithelium is closely dependent on the regulation of cell proliferation, differentiation, and apoptosis [4]. It seems that diet and its components have a profound influence on these processes, and its associated mechanism being little elucidated [2,4]. Our results show that bioactive compounds from mustard meal extract maintained cell viability, stimulated the proliferation of intestinal Caco-2/XTM-29, and counteracted the decreasing effect caused by *E. coli*-LPS in a manner similar to the ZnO treatment. Likewise, the treatment with emodin, a natural compound extracted from rhubarb, thuja, and other Chinese herbs, increased cell survival by slowing apoptosis of NCM460 human intestinal epithelial cells in a sepsis cellular model using LPS [25], whereas an aqueous extract of *Chlorella pyrenoidosa*, *Spirulina platensis*, and *Synechococcus* sp. PCC 7002 alleviated the production of intracellular ROS, depreciation of the plasma membrane, and the apoptosis induced by H_2_O_2_ [26].

However, the effects of bioactive compounds on cell viability and proliferation depend on their concentration and exposure time. Thus, [27] reported a concentration- and time-dependent effect of paeoniflorin (PF), a bioactive compound from *Paeonia lactiflora* Pallas plants on the viability of Caco-2 cells activated with LPS, which decreased significantly at 150 μM after 48 h of cultivation. The cellular viability was affected neither at 24 h nor after 48 h at concentrations lower than 150 μM [27]. Nevertheless, many cancer studies have shown that some plants rich in flavonoid polyphenols exhibit their antiproliferative effects. Studies by Dinicola et al. (2013) and Bagchi et al. (2000) found that extracts from Italia, Palieri, and Red Globe cultivars inhibited the growth of Caco-2 human colon cancer cells in a dose-dependent manner but enhanced the viability and growth of normal cells [28,29]. However, the anticancer effect of polyphenols from grape seed and others is based on their ability to reduce the accumulation of free reactive oxygen species (ROS). Powerful polyphenols from grape seed, such as pigallocatechin, anthocyanin, and gallic acid, were able, in an extract form, reduce the generation of ROS following LPS treatment in an IPEC-1 cell culture [22]. Moreover, procyanidin A2, a polyphenol isolated from grape seeds, demonstrated an inhibitory effect on LPS-induced ROS production in RAW264.7 macrophages [30]. In this regard, a large amount of data in the literature indicate that reactive oxygen species are the “second messenger” in the signaling pathway cascade that controls cell growth, proliferation, and apoptosis, as well as activates the mitogen-activated protein kinase (MAPK) cascades [28,31,32], especially JNK and p38-MAPK, which inhibit cell-cycle progression and promote apoptosis [33]. It was shown that JNK and p38-MAPK might be a pro-apoptotic signal in damaged intestinal epithelial cells.

Indeed, in our study, the exposure to LPS damaged epithelial cell membrane and induced mitochondrial membrane depolarization and apoptosis (Figure 3, Table 5). It is believed that alteration of the mitochondrial membrane potential leads to an increase in the mitochondrial permeability, release of cytochrome C in the cytosol, and triggers the apoptotic cascade [34]. The pretreatment of cells with mustard meal extract and ZnO solution counteracted the significantly exacerbated levels of ROS and mitochondrial depolarization. The mechanism involves the activation of mitogen-activated protein kinase pathways that regulate the cell cycle and other cell-proliferation-related proteins by ROS [32]. P-38 belongs to the MAPK family, which plays key roles in cell proliferation, differentiation, and apoptosis. It is activated by, among others, LPS, the main stimulator of ROS production [35]. Once activated, p-38 can induce cell-cycle arrest in the G0/G1 and S phases [35]. In our study, an overexpression of p38MAPK (2.28-fold change) was observed following the induction of LPS together with a decrease in the percentage of cells in the G0/G1 and S phases of the cell cycle, as well as an increase in apoptosis. A decrease in the expression of p38-MAPK to the control level was observed in the cell culture pretreated with mustard meal extract (1.04-fold change) and ZnO (1.22-fold change).

Epithelial integrity is another characteristic that is mandatory for a normal functioning epithelium. This integrity is maintained by tight cell–cell junctions that prevent the penetration of pathogens and toxins and stabilize the tissue [21]. In our study, the integrity of the in vitro Caco-2/HT29-MTX monolayer epithelium was evaluated by measuring the LDH, TEER, FD4, and TJ expression (both gene and protein expression). Firstly, we observed that the LDH assay, a marker of cellular membrane damage, showed a significant difference in the LDH release between the control cells and LPS-treated cells, suggesting that LPS is a strong inducer of membrane damage. Once the cell membrane is destroyed, the permeability is immediately affected. Indeed, measurements of the TEER and FD4, two of the most important indicators of barrier integrity, decreased dramatically after 6 h of LPS treatment, particularly for TEER, from 1127.2 Ω at 0 h to 831.0 Ω at 6 h (decreasing to 73%), compared to the control. Meanwhile the concentration of FD4, a marker of permeability, increased by 3.26-fold the basal compartment of coculture challenged with LPS, suggesting a significant disruption of the monolayer integrity, most likely through LPS-induced tight-junction protein damage. The disorder of TJs allows for the translocation of toxins (such as bacterial LPS) or other components from the intestinal lumen into the intestinal wall, causing endotoxemia [36] and changing the innate and acquired immune responses [5]. In the present study, TJ proteins (claudins, zonula-1, and occludin) registered a decreased in gene and protein expression following exposure to LPS, action, which was further increased by the ME extract and ZnO. This is very important if we take into consideration the importance of these proteins, which not only seal epithelial cells but, as recent studies show, are associated with different signaling proteins [37]. Conversely, genes coding for TLRs were significantly overexpressed by LPS. Indeed, TLR4, which specifically recognizes LPS [21], was the most overexpressed (6.5 fc), together with its co-receptor MD2 (4.3 fc). Their interaction triggered MyD88 (4.2fc), leading to the NF-kB pathway’s alteration. Under exposure to ME and ZnO, their expression returned to the control levels. Many studies report that dietary polyphenols play a key role in the improvement in epithelial membrane damage. In [36], the authors exhaustively explain in their 2023 article that among active dietary components, polyphenols can contribute to the maintenance of epithelial membrane integrity in several ways by acting primarily as prebiotics that stimulate the growth of beneficial bacteria and the production of short-chain fatty acids, thus inhibiting the translocation of pathogens. Secondly, polyphenols interfere in the up-/downregulation of tight-junction proteins and intracellular signaling molecules, such as NF-kB, MAPKs, and Nrf-2, as well as TLR-4, contributing to the stability and functionality of epithelial cells and the control of oxidative stress and inflammation [36]. Of great importance, a class of serine-threonine kinases (PKCs) is also involved in ensuring the stability and functionality of the intestinal barrier [38]. It is known, for example, that PKC can strengthen or weaken tight-junction proteins according to the cell type and the activation conditions [38]. For example, improvements in cell permeability, TEER, and cell shape were noticed in colonic HT 29 cells through the activation of PKC by phorbol ester PMA. Confocal microscopy highlighted clear changes in occluding and claudin-1 proteins. In contrast, the activation of PKC with the same PMA increased the permeability and disassembly of tight-junction proteins in canine kidney (MDCK I) cells [39]. PKC activation by bioactive compounds, such as oxyresveratrol (OXY), an isomer of hydroxylated resveratrol, from different plants increased TJ-related genes and proteins (claudin-1, occludin, and zonula-1) and reduced permeability in Caco-2 cells [40]. OXY also increased the expression levels of mitogen-activated protein kinase (MAPK) genes. It seems that PKC increased the integrity of junction proteins through transcription factors in the GATA family in the case of certain activators [38].

The results of our study suggest that the mustard meal extract, through its bioactive compounds, successfully suppressed the disruptive effects of LPS on epithelial integrity and permeability by inhibiting the overexpression of TLRs (1 to 10) and the signaling pathway molecules NF-kB and p-38 MAPK. PKC-A and PKC-B were also ameliorated by the mustard meal extract and ZnO, with comparable results between the two treatments.

## 4. Materials and Methods

### 4.1. Obtaining the Mustard Meal Extract and ZnO Solution

Five grams of mustard seed meal was mixed with 35 mL of extraction solvent (methanol 80%) at a sample–solvent ratio of 1:7 (*w*/*v*) and shaken continuously overnight. The supernatant was recovered by centrifugation at 4500 rpm for 20 min and filtered. To be used in cellular culture, the methanolic supernatant was concentrated with a rotary vacuum concentrator (CHRIST AVC 2-18CO Plus, Wertheim, Germany) to remove the methanol, and 1 mL of water was added to the remaining organic residue. The aqueous extract was sterile-filtered, diluted, and used in cell cocultures. The ZnO solution was prepared as described in [13].

### 4.2. Detection of Total Polyphenols and Antioxidant Activity

The total polyphenol content (TPC) of mustard seed meal was determined spectrophotometrically by the Folin–Ciocalteu method, as described in [41]. TPC was calculated from a calibration curve, using Gallic acid as standard. The results are expressed as mg gallic acid equivalents (mg GAE)/g. The antioxidant activity was measured using the ABTS 2,20-azinobis-(3-ethylbenzothiazoline-6-sulfonate) assay as described by Vlassa et al. [41].

### 4.3. Cell Coculture

Cell studies were carried out using a coculture of Caco-2 epithelial cells (American Type Culture Collection, Manassas, VA, USA) and HT29-MTX (European Collection of Authenticated Cell Cultures-ECACC, Salisbury, UK) mucus-secreting cells. The Caco-2 and HT29-MTX cells represent the two most abundant epithelial cells (absorptive and mucin-producing goblet cells, respectively) found in the intestinal epithelium [38,42,43]. Cells were grown in 75 cm^2^ flasks in Minimum Essential Medium Eagle (Caco-2) and RPMI-1640 (HT29-MXT) medium (Sigma Aldrich, St. Louis, MO, USA), both supplemented with 10% (*v*/*v*) of heat inactivated fetal bovine serum (Sigma Aldrich, St. Louis, MO, USA), 2 mM L-glutamine (Sigma Aldrich, St. Louis, MO, USA), and 1% antibiotic–antimycotic (Sigma Aldrich, St. Louis, MO, USA) at 37 °C in a 5% CO_2_ humidified atmosphere. For the HT29-MTX culture, 1% Non-Essential Amino Acid Solution (Sigma Aldrich, St. Louis, MO, USA) was added to the culture medium. The culture medium was changed every three days. For the coculture experiments, the cells were seeded at a ratio of 9 (Caco-2) to 1 (HT29-MXT) and grown in MEM, as described in [13].

### 4.4. Assessment of Epithelial Renewal

#### 4.4.1. Coculture Cell Viability Measurement (MTT Assay)

Cells (Caco-2 and HT29-MTX) were seeded at a 9:10 ratio and maintained in 96-well flat-bottomed microplates (Corning, Sigma-Aldrich, St. Louis, MO, USA) at a concentration of 1 × 10^5^ cells/100 μL in culture media (MEM + 1% nonessential amino acids) until reaching 80% confluence (2 days). Then, the coculture was treated with ME (dilution 1/50) and ZnO (50 µM) for 2 h and challenged with *E. coli*-LPS (serotype O26:B6, 5 μg/mL) for another 4 h. Cell viability was assessed using the MTT [3-(4,5-dimethylthiazol-2-yl)-2,5-diphenyl tetrazolium bromide] assay, according to the manufacturer’s instructions, after the incubation period elapsed, as described in [44]. The absorbance was measured at 450 nm using an ELISA microplate reader (Tecan Infinite M200 PRO, Viena Austria). The cellular viability is expressed as the percent of the control, which is considered 100%. Three independent experiments were performed for all tests.

#### 4.4.2. Coculture Cell Proliferation Measurement

Cells (Caco-2 and HT29-MTX) were seeded at a 9:10 ratio and maintained in 6-well microplates (Corning, Sigma Aldrich, St. Louis, MO, USA) at a concentration of 1 × 10^5^ cells/well in culture media (MEM + 1% nonessential amino acids) until reaching 80% confluence (2 days). Then, the coculture was treated with ME (dilution 1/50) and ZnO (50µM) for 2 h and challenged with *E. coli*-LPS (5 μg/mL) for another 4 h. The cell proliferation was measured based on the Ki67 expression using the Muse^®^ Ki67 Proliferation Kit (Luminex Corporation, Austin, TX, USA), according to the manufacturer’s recommendations. Briefly, cells were harvested, washed with PBS, fixed, and then permeabilized via incubation for 15 min at room temperature. Then, the cells were stained with 10 μL of either Muse^®^ Hu IgG1-PE or Muse^®^ Hu Ki67-PE, incubated for 30 min at room temperature, and the percentages of the Ki67-positive cell populations in the control and treated cells were measured using the Guava Muse™ Cell Analyzer (Luminex Corporation, Austin, TX, USA). Three independent experiments were performed for all tests.

#### 4.4.3. Detection of Apoptosis

Apoptosis, in constant balance with cell proliferation, was measured by flow cytometry using the Muse^®^ Annexin V & Dead Cell Kit (Luminex Corporation, Austin, TX, USA) and the Guava^®^ Muse^®^ Cell Analyzer for the acquisition of live, early-apoptotic, late-apoptotic, and dead cells. The cytometry test was performed after the cultured Caco-2/HT-29 MTX cells (9:10) reached 80% confluence, were treated with mustard meal extract (1/50) and ZnO (50 µM) for 2 h, and then challenged with *E. coli*-LPS 5 μg/mL for another 4 h. Three independent experiments were performed for all tests.

#### 4.4.4. Detection of the Cell Cycle

Cells (Caco-2 and HT29-MTX) were seeded at a 9:10 ratio and maintained in 6-well microplates (Corning, Sigma Aldrich, St. Louis, MO, USA) at a concentration of 1 × 10^5^ cells/well in culture media (MEM + 1% nonessential amino acids) until reaching 80% confluence (2 days). Then, the coculture was treated with ME (dilution 1/50) and ZnO (50 µM) for 2 h and then challenged with *E. coli*-LPS (5 μg/mL) for another 4 h. The percentages of cells in the G0/G1, S, and G2/M phases of the cell cycle were measured using the Muse^®^ Cell Cycle Kit (Luminex Corporation, Austin, TX, USA), based on nuclear DNA intercalating stain propidium iodide (PI) in the presence of RNAse A, according to the manufacturer’s recommendations. Briefly, cells were harvested, washed with PBS, and then fixed with 70% ice-cold ethanol solution by overnight incubation at −20 °C. Then, the cells were stained with 200 μL of Muse^®^ Cell Cycle Reagent, incubated for 30 min at room temperature in the dark, and the fluorescence intensity of the PI measured using the Guava Muse™ Cell Analyzer (Luminex Corporation, Austin, TX, USA). The percentages of treated or nontreated cells in each cell-cycle phase (G0/G1, S, and G2/M) were determined. Three independent experiments were performed for all tests.

### 4.5. Assessment of the Monolayer Integrity

#### 4.5.1. Measurement of the Trans-Epithelial Electrical Resistance (TEER)

Caco-2/HT29-MTX cells were seeded at a ratio of 9:10 in a concentration of 1 × 10^6^ cells in 24-well transparent PET membrane inserts (Falcon, Corning, Durham, NC, USA) with a 0.4 µm pores size. The cells were cultured in MEM + 1% nonessential amino acids changed every 2 days until complete differentiation was achieved (21 days). When the differentiation was complete, the cocultured cells were treated with ME (dilution 1/50) and ZnO (50 μM) and incubated for 2 h. Then, the cells were challenged with 5 μg/mL *E. coli*-LPS and incubated for another 4 h. The TEER was measured at 0, 6, 24, 48, and 24 h using Millicell ERS-2 Voltohmmeter (Millipore, Merck, Darmstadt, Germany). Experimental TEER values were expressed as kOhms × cm^2^. All experiments were performed in three independent experiments.

#### 4.5.2. Measurement of Tight-Junction Proteins (Gene Expression)

The gene and protein expression of tight-junction proteins (TJs); claudins (CLDN-1, -2, -4, -5, -14, -20, and -23; occludin (OCCLDN); and zonula 1 (ZO1) was detected by qPCR array after the incubation of cocultured Caco2/HT-MTX (9:10) with ME (1/50) and ZnO (50 μM) for 2 h and *E. coli*-LPS (5 μg/mL) for 4 h, as described in [13]. For the qPCR array, the total RNA (tRNA) was extracted using the Qiagen RNeasy mini-kit (QIAGEN GmbH, Hilden, Germany), following the manufacturer’s instructions, and the complementary DNA (cDNA) was synthetized using the M-MuLV reverse transcriptase kit (Thermo Fisher Scientific, Waltham, MA, USA). The gene expression coding for tight-junction proteins were performed using synthetized cDNA, primers for specific gene sequences, qPCR SYBER Green/Fluorescein Master Mix 2X, and Rotor-Gene-Q Pure Detection (QIAGEN GmbH, Germany), as described in [45]. A negative control, including PCR mix without cDNA, was also used. The qPCR cycling was performed as described in [46]. The PCR data were normalized using the expression of two housekeeping (*ACTB* and *GAPDH*) genes, which were selected for their stability from a panel of five genes using the NormFinder Excel-based software v21, 2024. The results were analyzed using the 2^(−ΔΔCT)^ method [47,48], and are expressed as the relative fold change (Fc) compared to the untreated (control) cells. The qPCR analyses were performed in triplicate. To validate the gene profiles provided by the qPCR array, the protein expression level of claudin-4 (CLDN4) was measured by immunofluorescence in all samples used in the qPCR array. The Caco-2 cells and HT-29 cells (9:1 ratio) were grown on cell imaging slides (Eppendorf GmbH, Hamburg, Germany) at a concentration of 1 × 10^5^ cells/mL for 7 days after reaching confluence. Then, the coculture was treated for 2 h with ME (dilution 1/50) and ZnO (50µM) and challenged with *E. coli*-LPS (5 μg/mL) for another 4 h; all incubation steps were performed at 37 °C in a 5% CO_2_ humidified atmosphere. At the end of incubation, the cells were washed 3 times with PBS (phosphate-buffered saline), fixed with 100% methanol (10 min at −20 °C), and blocked with a mixture of 10% FBS (fetal bovine serum) and 10% GS (goat serum) in TBS (Tris-buffered saline) for 30 min at room temperature. After blocking, the samples were incubated (overnight, at 4 °C) with anti-Claudin-4 antibody (Abcam, Cambridge, UK) at a dilution of 1:50. Slides were washed with TBS and incubated with secondary fluorescent-conjugated antibodies (CF594 antibody, Sigma-Aldrich Chemie GmbH, Schnelldorf, Germany) at a dilution of 1/50. For nuclear staining, 1 µg/mL Hoechst 33, 342 (Cell Signaling Technology, Inc, Danvers, MA, USA) was added during the last 15 min of incubation with the secondary antibody. Images were captured using a Nikon Ts2RFL inverted microscope and a CCD camera and processed using ZEN 3.2 (Blue edition) (Carl Zeiss Microscopy GmbH, Oberkochen, Germany) or ImageJ (Fiji version) software 1.64. Three independent experiments were performed.

#### 4.5.3. Measurement of the Cellular Permeability

Caco-2/HT29-MTX cells were cultured at a ratio of 9:10 and a concentration of 1 × 10^6^ cells in 24-well transparent PET membranes inserts (Falcon, Corning, Durham, NC, USA) with a 0.4 µm pore size. The medium used was MEM + 1% nonessential amino acids, which was changed every 2 days. At 21 days after seeding, the cells were treated with ME (dilution 1/50) and ZnO (50μM) and incubated for 2 h. Then, the cells were challenged with 5 μg/mL *E. coli*-LPS and incubated for another 6 h. The paracellular permeability of the cellular monolayers was determined by the transport of fluorescein isothiocyanate (FITC)-conjugated dextran 4 kDa (FD4) dissolved in PBS to a final concentration of 1mg/mL, as described in [49]. Briefly, 100 µL of FD4 solution was added to the apical pole of the transwells containing the overtreated or untreated cells and incubated for 2 h at 37 °C in the dark. To measure the passage of FD4 across the monolayer, samples of the supernatant from the apical and basal poles of the transwells were collected, and the fluorescence of FD4 was measured at a 490 nm excitation and 520 nm emission using a Varioskan™ LUX multimode microplate reader (Thermo Fisher Scientific, Waltham, MA, USA). FD4 is expressed as μg/mL, using a calibration curve of the fluorescence intensity of the various FD4 concentrations.

#### 4.5.4. Monolayer Damage Detection (LDH Assay)

Lactate dehydrogenase (LDH) activity in the coculture supernatant was detected according to the kit’s instructions provided by Sigma-Aldrich (Sigma Aldrich, St. Louis, MO, USA). The Caco2/HT-MTX (9:10) cocultures were treated with ME 1/50 and ZnO (50 μM) for 2 h and *E. coli*-LPS (5 μg/mL) for another 4 h. Briefly, equal volumes (25 µL) of cell coculture supernatant and assay buffer were mixed in 96-well plates and then with 50 µL of substrate mix diluted in assay buffer. The plate was further incubated at 37 °C, and the absorbance was measured every 30 s until the highest absorbance value of the samples exceeded the value of the highest standard (12.5 nmol/well). The LDH activity was calculated according to the kit’s instructions. The absorbance measurements were performed after 2–3 min using a Varioskan™ LUX multimode microplate reader (Thermo Fisher Scientific, Waltham, MA, USA).

#### 4.5.5. Measurement of the Innate Immunity Parameters (Genes and Expression)

The gene and protein expressions of Toll-like receptors (TLR-2, -3, -4, -6, -8, -9, and -10), key markers of the innate immune response in the epithelium, were detected using a qPCR array from lysates of Caco2/HT-MTX (9:10) cells cultured for 4 days. Cells were then treated with ME (1/50) and ZnO (50 μM) for 2 h and LPS (5 μg/mL) for another 4 h, as described for the tight-junction proteins. The sequences of primer pairs used in this study are provided in Appendix A.

#### 4.5.6. Measurement of the Signaling Molecules (Gene Expression)

Several key signaling and adapter molecules (NF-kB, TAK1, p-38a MAPK, PKC-A, PKC-B, MyD88, MD-2, TRAF6, IRAK1, etc.) involved in the renewal, integrity, and immune functions of intestinal epithelia were assessed using qPCR arrays and as described for the tight-junction proteins. The sequences of the primer pairs used in this study are provided in Appendix A.

#### 4.5.7. Mitochondrial Morphology Assay of the Live Cells

Mito-Lite Red FX600 (AAT Bioquest, Pleasanton, CA, USA), a set of fluorogenic probes that label the mitochondria of live cells, was used to evaluate the morphology and cell viability of the mitochondria. The mitochondrial dyes accumulate in the mitochondria via the mitochondrial membrane potential gradient. Cells (Caco-2 and HT29-MTX) were seeded at a 9:10 ratio and maintained in 24-well microplates (Corning, Sigma Aldrich, St. Louis, MO, USA) at a concentration of 1 × 10^5^ cells/100 µL of well in culture media (MEM + 1% nonessential amino acids) until reaching 100% confluence (3 days). Then, the coculture was treated with ME (dilution 1/50) and ZnO (50 µM) for 2 h and challenged with *E. coli*-LPS (5 μg/mL) for another 4 h. At the end of incubation, the culture medium was removed, and the cells were washed with PBS. Further, cells were incubated with 2 μL of 500 × MitoLite stock solution in 1000 μL of PBS for 2 h at 37 °C. Then, they were washed with prewarmed PBS, and the fluorescence intensity was observed using a Nikon ECLIPSE Ts2RFL fluorescence microscope (New York, NY, USA) with a Cy5 HYQ red excitation filter. Three independent experiments were performed.

#### 4.5.8. Mitochondrial Membrane Potential

Changes in the mitochondria, especially in the depolarization of the internal mitochondrial membrane potential, are distinctive markers of mitochondrial dysfunction and cellular stress. In the present study, the effect of *E. coli* LPS and of ME and ZnO treatments on the mitochondrial membrane potential depolarization was assessed by flow cytometry using the Muse™ MitoPotential Kit and Guava Muse™ Cell Analyzer (Luminex Corporation, Austin, TX, USA). It is expressed as the percentage of live, depolarized/dead, and dead cells, according to the manufacturer’s recommendations. The cells (Caco-2 and HT29-MTX) were cultured in a manner similar to that for the mitochondrial morphology. They were incubated with Muse™ MitoPotential working solution (dilution 1:1000 with 1X assay buffer) at 37 °C for 20 min. Then, 5 μL of Muse™ 7-AAD was added, incubated at RT for 5 min, and the fluorescence intensity of the MitoPotential dye was measured using the Guava Muse™ Cell Analyzer (Luminex Corporation, Austin, TX, USA).

#### 4.5.9. Statistical Analysis

The results are presented as the mean ± standard error of the mean. StatView software 6.0 (SAS Institute, Cary, NC, USA) with one-way ANOVAs and Student’s *t*-tests were used to compare the statistical differences between the values of all parameters used following the experimental treatments. In addition, Fisher’s least square difference procedure was used, and *p*-values < 0.05 are considered significant.

## 5. Conclusions

Our results demonstrate that LPS not only is a distinct promoter of inflammation but also has a pronounced effect on the functions of the intestinal epithelial monolayer, such as renewal and integrity. It decreased cell proliferation, increased apoptosis, and arrested the cell cycle in the G0/G1 phases. LPS altered the epithelium’s permeability by decreasing the transepithelial electrical resistance and tight-junction protein expression, as well as by increasing the activity of LDH and the expression of Toll-like receptors. The mechanism by which LPS induces all these disturbances involved PKC, p38 MAPK, and NF-κB signaling molecules which were overexpressed. Our results also show that the extract derived from mustard byproduct is a rich, complex matrix that is able to suppress the effects of LPS. There were no significant differences between the ME and ZnO, suggesting that it could be an alternative source to replace the use of in-feed ZnO or other antimicrobials (e.g., antibiotics). This study is a preliminary investigation that used a simplified in vitro cellular model, and the findings must be confirmed by in vivo tests.

## Figures and Tables

**Figure 1 ijms-26-00273-f001:**
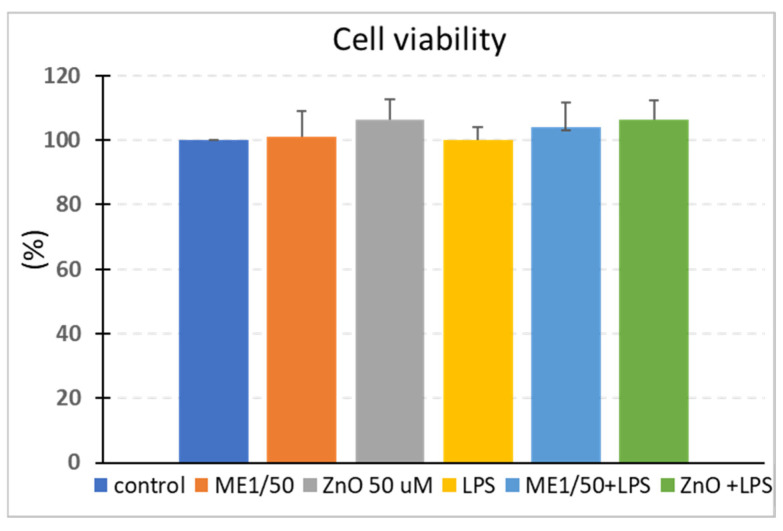
Effects of the experimental treatments on the viability of Caco-2/HT29-MTX cells. Caco-2/MTX cells that had achieved 80% confluence were preincubated for 2 h at 37 °C in the presence of mustard extract (ME) and zinc oxide (ZnO) and then exposed to LPS for another 4 h. The cell viability was determined using an MTT test. ANOVA tests were performed to compare the viability of the control and treated cells. The results are expressed as the percentage of control cells representing the mean ± SEM of three independent experiments. Control = untreated cells; ME = cells treated for 2 h with mustard meal extract (1/50 dilution); ZnO = cells treated for 2 h with ZnO (50 µM); LPS = cells treated for 4 h with 5 µg LPS/mL; ME+LPS = cells treated for 2 h with ME and then challenged with LPS for 4 h; ZnO+LPS = cells treated for 2 h with ZnO and then challenged with LPS for 4 h.

**Figure 2 ijms-26-00273-f002:**
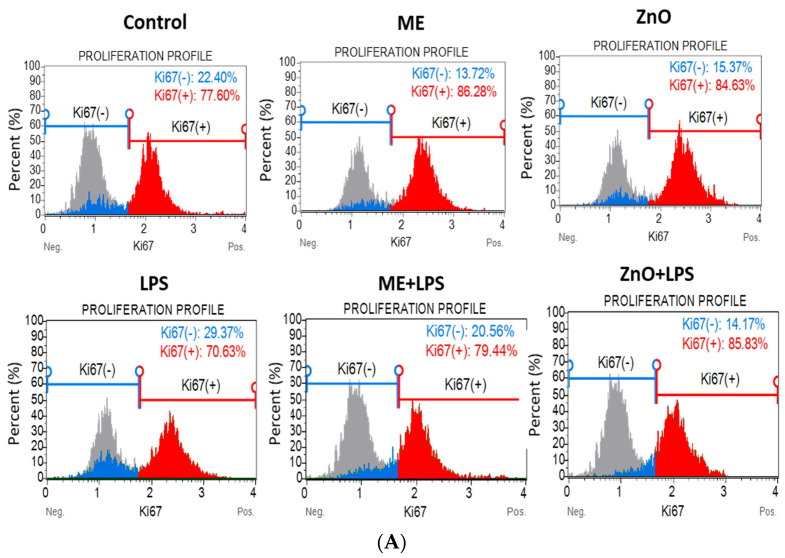
Effects of the ZnO and ME extract on the proliferation of Caco-2/HT29-MTX cells challenged with LPS. Cell proliferation was measured on the basis of the expression of Ki67 using the Muse^®^ Ki67 Proliferation Kit (Luminex Corporation, Austin, TX, USA) and the Guava Muse™ Cell Analyzer. (**A**) Representative flow cytometry assay showing the staining distribution among the different treatments, Ki67(−) grey and Ki67(+)-red; (**B**) quantification of the Ki67(-) and ROS(+) cells. ^a,b,c,d,e,f^ Histograms of each treatment with different superscript letters are significantly different (*p* < 0.050). The results are expressed as the mean ± SEM of three independent experiments. Control = untreated cells; ME = cells treated for 2 h with mustard meal extract (1/50 dilution); ZnO = cells treated for 2 h with ZnO (50 µM); LPS = cells treated for 4 h with 5 µg LPS/mL; ME+LPS = cells treated for 2 h with ME and then challenged with LPS for 4 h; ZnO+LPS = cells treated for 2 h with ZnO and then challenged with LPS for 4 h.

**Figure 3 ijms-26-00273-f003:**
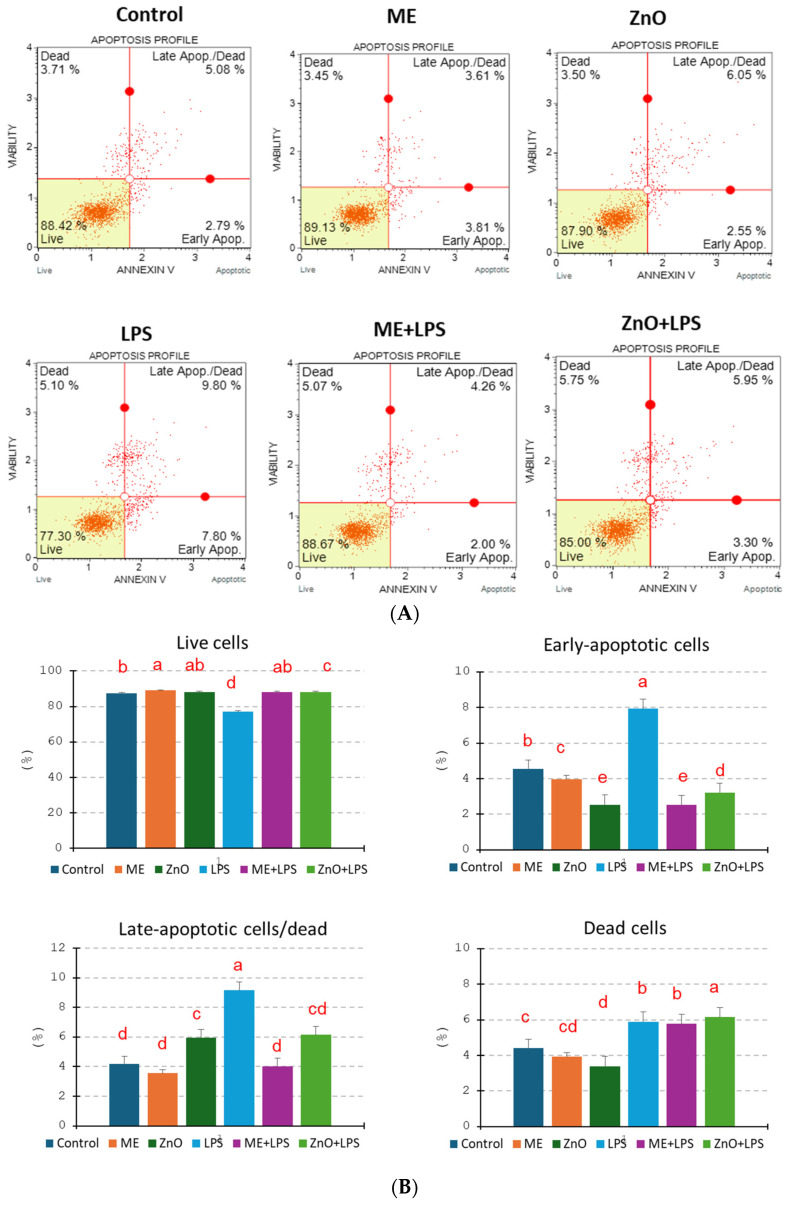
Effects of the ZnO and ME extract on the apoptosis of Caco-2/HT29MTX cells challenged with LPS. The apoptosis profiles were obtained using the Muse Annexin & Dead Cell kit and the Guava Muse™ Cell Analyzer and they are expressed as the percentages of live, early-apoptotic, late-apoptotic, and dead cells of the total cells. The results are expressed as the mean ± SEM for three independent experiments: (**A**) representative dot plots sorted by flow cytometry; (**B**) quantification of the apoptotic cells corresponding to each treatment. ^a,b,c,d,e^ Histograms for each treatment with different superscript letters are significantly different (*p* < 0.050). Control = untreated cells; ME = cells treated for 2 h with the mustard meal extract (1/50 dilution); ZnO = cells treated for 2 h with ZnO (50 µM); LPS = cells treated for 4 h with 5 µg LPS/mL; ME+LPS = cells treated for 2 h with ME and then challenged with LPS for 4 h; ZnO+LPS = cells treated for 2 h with ZnO and then challenged with LPS for 4 h.

**Figure 4 ijms-26-00273-f004:**
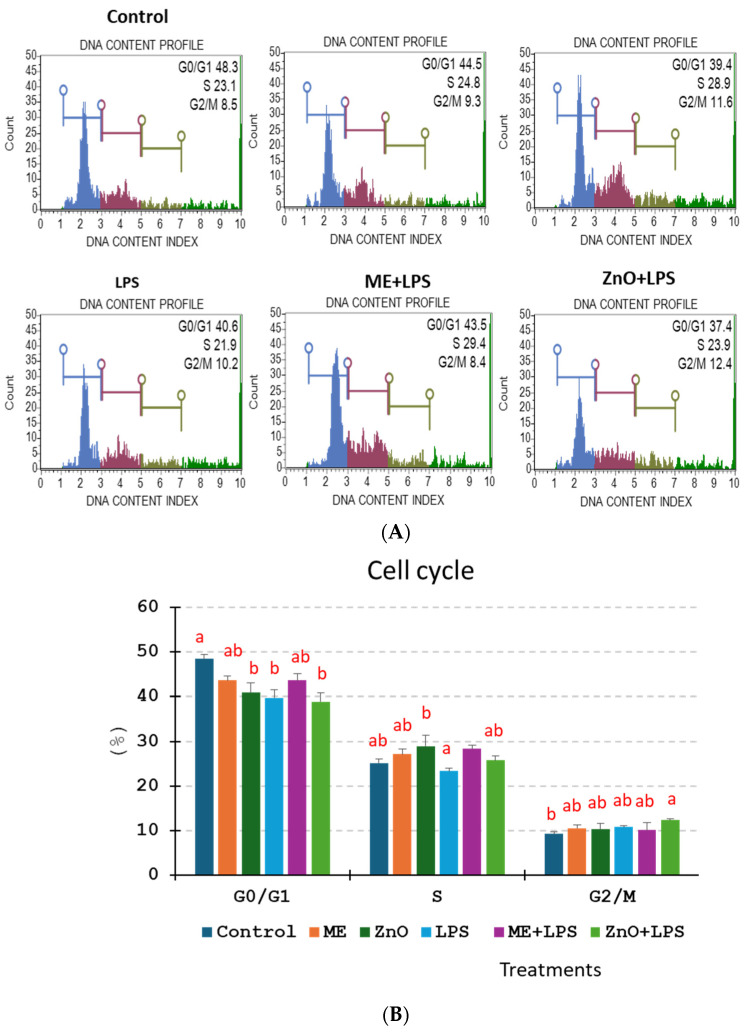
Effect of the ZnO and ME extract on the cell cycle of Caco-2/HT29-MTX cells challenged with LPS. Phases of cell cycle (G0/G1; S; G2/M) was analyzed by using Muse^®^ Cell Cycle Kit and the Guava Muse™ Cell Analyzer, according to manufacturer’s recommendations. The results are expressed as the mean ± SEM for three independent experiments. (**A**) Representative flow cytometry assay showing staining distribution among the different treatments and (**B**) Percentage of cells in the G0/G1 (blue), S (red), G2/M (dark green) phases of the cell cycle. ^a,b^ Histograms for each treatment with different superscript letters were significantly different (*p* < 0.050). Control = untreated cells; ME = cells treated for 2 h with mustard meal extract (1/50 dilution); ZnO = cells treated for 2 h with ZnO (50 µM); LPS = cells treated for 4 h with 5 µg LPS/mL; ME+LPS = cells treated for 2 h with ME and then challenged with LPS for 4 h; ZnO+LPS = cells treated for 2h with ZnO and then challenged with LPS for 4 h.

**Figure 5 ijms-26-00273-f005:**
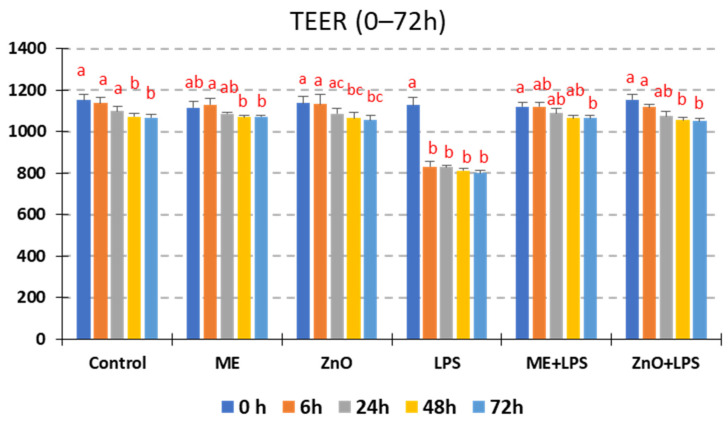
Modulation of the trans-epithelial electrical resistance (TEER) in Caco2/HT29-MTX in response to LPS challenge and ME and ZnO treatments. Caco-2 and HT29-MTX were seeded at a 9:10 ratio and maintained in 24-well microplates at a concentration of 1 × 10^5^ cells/well in culture media until complete differentiation was achieved (21 days). Then, the coculture was treated with ME (dilution 1/50) and ZnO (50 µM) for 2 h and challenged with *E. coli*-LPS (5 μg/mL) for another 4h. The TEERs measured at 0, 6, 24, 48, and 72 h decreased dramatically after 6 h of exposure to LPS. The results are expressed as the mean ± SD, n = 6. ^a,b,c^ Histograms for each treatment with different superscript letters are significantly different (*p* < 0.050). Control = untreated cells; ME = cells treated for 2 h with mustard meal extract (1/50 dilution); ZnO = cells treated for 2 h with ZnO (50 µM); LPS = cells treated for 4 h with 5 µg LPS/mL; ME+LPS = cells treated for 2 h with ME and then challenged with LPS for 4 h; ZnO+LPS = cells treated for 2 h with ZnO and then challenged with LPS for 4 h.

**Figure 6 ijms-26-00273-f006:**
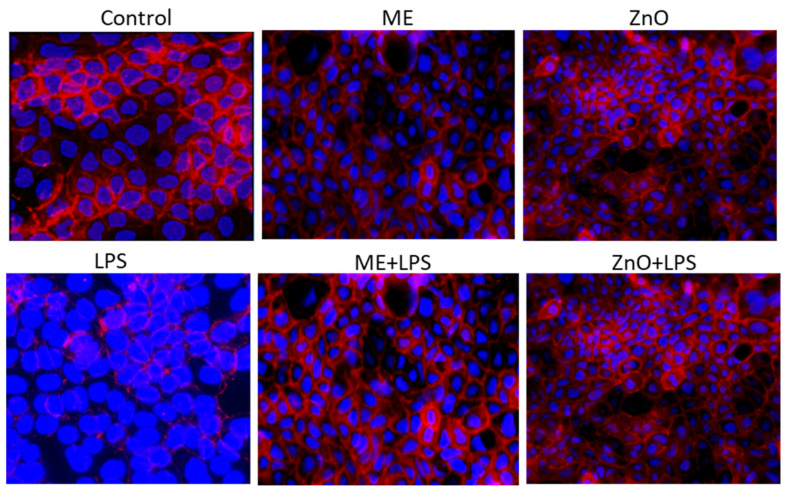
Protein expression of claudin-4 (CLDN4) measured by immunofluorescence. Cocultures of Caco-2 cells and HT-29 cells (9:1 ratio) were grown on cell imaging slides at a concentration of 1 × 10^5^ cells/mL for 7 days after reaching confluence. Then, the coculture was treated for 2 h with ME (dilution 1/50) and ZnO (50µM) for 2 h and challenged with *E. coli*-LPS (5 μg/mL) for another 4 h. Fluorescence images were captured using a Nikon Ts2RFL inverted microscope, Amstelveen, The Netherlands (magnification 40 X) and a CCD camera and processed using ZEN 3.2 (Blue edition) (Carl Zeiss Microscopy GmbH, Oberkochen, Germany) or ImageJ (Fiji version 1.64) software. Three independent experiments were performed.

**Figure 7 ijms-26-00273-f007:**
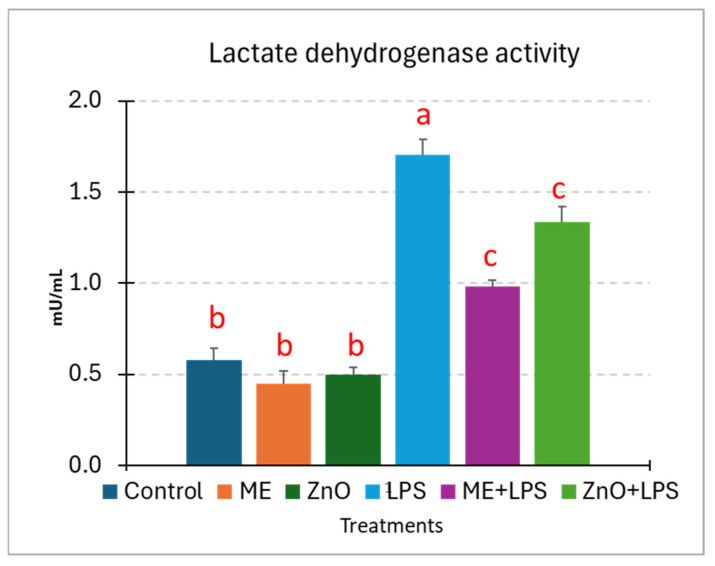
Membrane damage assessed by LDH in Caco2/HT 29-MTX cells in response to the LPS challenge and ME and ZnO treatments. Caco-2 and HT29-MTX cells were seeded at a 9:10 ratio and maintained in 96-well microplates at a concentration of 1 × 10^5^ cells/well in culture media until complete differentiation was reached. Then, the coculture was treated with ME (dilution 1/50) and ZnO (50 µM) for 2 h and challenged with *E. coli*-LPS (5 μg/mL) for another 4 h. The results are expressed as the mean ± SD, n = 6. ^a,b,c^ Histograms for each treatment with different superscript letters are significantly different (*p* < 0.050). Control = untreated cells; ME = cells treated for 2 h with mustard meal extract (1/50 dilution); ZnO = cells treated for 2 h with ZnO (50 µM); LPS = cells treated for 4 h with 5 µg LPS/mL; ME+LPS = cells treated for 2 h with ME and then challenged with LPS for 4 h; ZnO+LPS = cells treated for 2 h with ZnO and then challenged with LPS for 4 h.

**Figure 8 ijms-26-00273-f008:**
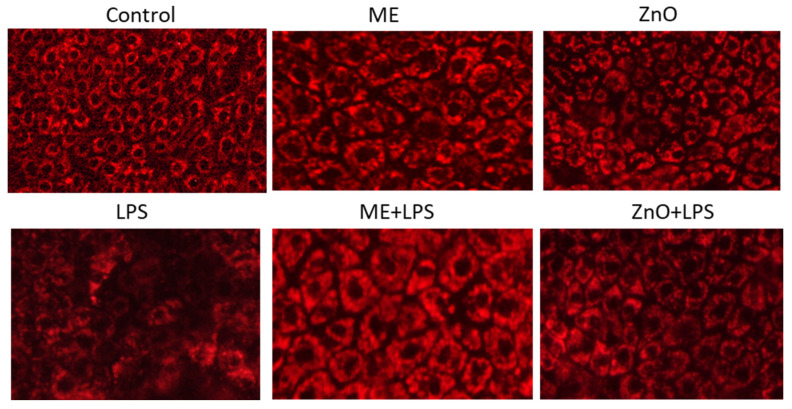
Fluorescence images of Caco2/HT29-MTX cells stained with the mitochondria dye MitoLite™ Red FX600 after exposure to LPS and the ME and ZnO experimental treatments. Cells (Caco-2 and HT29-MTX) were seeded at a ratio of 9:10 and maintained on 24-well microplates at a concentration of 1 × 10^5^ cells/well in culture media until reaching confluence. Then, the coculture was treated with ME (dilution 1/50) and ZnO (50 µM) for 2h and challenged with *E. coli*-LPS (5 μg/mL) for another 4 h. Fluorescence images of Caco2/HT29-MTX cells stained with the mitochondria dye MitoLite™ Red FX600 were captured using a fluorescence microscope with a Cy5 HYQ red excitation filter.

**Table 1 ijms-26-00273-t001:** Effects of ME and ZnO on genes coding for tight-junction proteins in LPS-challenged Caco2/HT-29 cocultured cells.

Functional Classes of Genes	Gene	Experimental Treatment *												
		Control	LPS		ME		ME+LPS			ZnO		ZnO+LPS		
		Mean ± SEM **	Mean ± SEM **	Compared to Control ^#^	Mean ± SEM **	Compared to Control ^#^	Mean ± SEM **	Compared to Control ^#^	Compared to LPS ^#^	Mean ± SEM **	Compared to Control ^#^	Mean ± SEM **	Compared to Control ^#^	Compared to LPS ^#^
Tight junction proteins	*CLDN1*	1.0 ± 0.0 ^b^	0.3 ± 0.0 ^c^	down	1.7 ± 0.1 ^a^	up	2.4 ± 0.4 ^a^	up	up	3.3 ± 0.3 ^a^	up	1.9 ± 0.2 ^b^	up	up
	*CLDN2*	1.0 ± 0.0 ^b^	0.3 ± 0.1 ^c^	down	1.6 ± 0.1 ^a^	up	1.3 ± 0.1^b^	-	up	1.4 ± 0.1 ^b^	-	1.2 ± 0.3 ^b^	-	up
	*CLDN4*	1.0 ± 0.0 ^a^	0.6 ± 0.1 ^b^	down	1.3 ± 0.4 ^a^	-	1.2 ± 0.1 ^a^	-	up	1.5 ± 0.3 ^a^	-	1.2 ± 0.3 ^a^	-	up
	*CLDN5*	1.0 ± 0.0 ^a^	0.5 ± 0.1 ^b^	down	1.3 ± 0.3 ^a^	-	1.2 ± 0.1 ^a^	-	up	1.5 ± 0.4 ^a^	-	1.2 ± 0.3 ^c^	-	up
	*CLDN14*	1.0 ± 0.0 ^c^	0.6 ± 0.1 ^b^	down	1.8 ± 0.4 ^a^	up	1.7 ± 0.2 ^a^	-	up	1.0 ± 0.1 ^c^	-	1.5 ± 0.2 ^b^	-	up
	*CLDN20*	1.0 ± 0.0 ^a^	0.4 ± 0.1 ^b^	down	1.3 ± 0.1 ^a^	-	1.4 ± 0.1 ^a^	-	up	1.6 ± 0.4 ^a^	-	1.2 ± 0.3 ^b^	-	up
	*CLDN23*	1.0 ± 0.0 ^a^	0.4 ± 0.2 ^b^	down	1.2 ± 0.1 ^a^	-	1.1 ± 0.1 ^a^	-	up	1.1 ± 0.1 ^a^	-	1.0 ± 0.1 ^b^	-	up
	*OCCL*	1.0 ± 0.0 ^a^	0.5 ± 0.1 ^b^	down	1.2 ± 0.1 ^a^	-	1.1 ± 0.3 ^a^	-	up	1.1 ± 0.2 ^a^	-	1.5 ± 0.1 ^b^	up	up
	*ZO-1*	1.0 ± 0.0 ^a^	0.4 ± 0.1 ^b^	down	1.1 ± 0.2 ^a^	-	0.9 ± 0.2 ^ab^	-	-	0.8 ± 0.1 ^a^	-	0.9 ± 0.1 ^ab^	-	-
Regulators of tight junctions	*JAM-A*	1.0 ± 0.0 ^a^	0.5 ± 0.1 ^c^	down	1.2 ± 0.0 ^a^	-	0.5 ± 0.1 ^c^	down	-	1.6 ± 0.1 ^a^	-	1.4 ± 0.3 ^a^	-	up
	*MAGI2*	1.0 ± 0.0 ^a^	0.4 ± 0.1 ^c^	down	1.1 ± 0.2 ^a^	-	0.5 ± 0.1 ^c^	down	-	1.2 ± 0.1 ^a^	-	0.9 ± 0.1 ^c^	-	-
	*GNAI2*	1.0 ± 0.0 ^b^	0.7 ± 0.2 ^b^	-	1.4 ± 0.1 ^b^	-	1.3 ± 0.2 ^b^	-	-	3.9 ± 0.2 ^a^	up	3.5 ± 0.3 ^a^	up	up
	*MIO9B*	1.0 ± 0.0 ^a^	0.5 ± 0.1 ^b^	down	1.7 ± 0.5 ^b^	-	1.3 ± 0.1 ^a^	-	up	1.0 ± 0.2 ^b^	-	1.3 ± 0.1 ^a^	-	up
	*MLCK*	1.0 ± 0.0 ^a^	0.2 ± 0.0 ^b^	down	1.1 ± 0.1 ^b^	-	0.7 ± 0.1 ^a^	-	up	1.5 ± 0.1 ^a^	-	1.4 ± 0.2 ^a^	-	up
	*PTPN2*	1.0 ± 0.0 ^a^	0.4 ± 0.1 ^b^	down	0.8 ± 0.1 ^a^	-	0.9 ± 0.2 ^a^	-	up	0.8 ± 0.1 ^a^	-	1.0 ± 0.1 ^a^	-	up

* Cocultured Caco2/HT-29MTX cells were pretreated with ME and ZnO for 2 h and challenged with LPS, as described in Section 4 (Materials and Methods). At the end of the experiment, the cells were collected, and the gene expression was quantified by a qPCR array. ** The results are presented as the mean ± SEM. ^#^ The effects of the experimental treatments on gene expression were analyzed by one-way ANOVA followed by Fisher’s test; *p* < 0.05 is considered statistically significant. ^a,b,c^ Groups with different superscript letters are significantly different (*p* < 0.05), where “a” = the highest value. Up = upregulation; down = downregulation; - = no effect.

**Table 2 ijms-26-00273-t002:** Effects of ME and ZnO on the monolayer permeability of Caco2/HT29-MXT after challenge with LPS.

	Control	ME	ZnO	LPS	ME+LPS	ZnO+LPS	SEM
FD4 apical	5244.92 a	5124.71 a	5209.50 a	2015.29 b	5235.42 a	4745.96 a	365.50
FD4 basal	2019.83 a	2233.25 a	2940.10 a	4182.08 b	2665.00 a	2284.00 a	310.81

Values correspond to the mean ± SEM. Means without a common letter between treatments are significantly different (*p* < 0.05), Student’s *t*-test.

**Table 3 ijms-26-00273-t003:** The effects of ME and ZnO on the gene expression of Toll-like receptors (TLRs) in LPS-challenged Caco2/HT-29 cocultured cells.

Gene	Experimental Treatment *												
	Control	LPS		ME		ME+LPS			ZnO		ZnO+LPS		
	Mean ± SEM **	Mean ± SEM **	Compared to Control ^#^	Mean ± SEM **	Compared to Control ^#^	Mean ± SEM **	Compared to Control ^#^	Compared to LPS ^#^	Mean ± SEM **	Compared to Control ^#^	Mean ± SEM **	Compared to Control ^#^	Compared to LPS ^#^
*TLR1*	1.0 ± 0.0 ^b^	3.2 ± 0.2 ^a^	up	1.0 ± 0.1 ^b^	-	0.9 ± 0.1 ^b^	-	down	0.8 ± 0.1 ^b^	-	1.4 ± 0.3 ^b^	-	down
*TLR2*	1.0 ± 0.0 ^b^	1.8 ± 0.1 ^a^	up	0.8 ± 0.2 ^b^	-	0.5 ± 0.1 ^c^	down	down	0.8 ± 0.2 ^b^	-	0.5 ± 0.1 ^c^	down	down
*TLR3*	1.0 ± 0.0 ^a^	1.5 ± 0.1 ^a^	-	1.1 ± 0.2 ^a^	-	1.4 ± 0.2 ^a^	-	-	1.1 ± 0.2 ^a^	-	1.2 ± 0.3 ^a^	-	-
*TLR4*	1.0 ± 0.0 ^b^	6.5 ± 0.5 ^a^	up	0.9 ± 0.0 ^b^	-	0.8 ± 0.1 ^b^	-	down	0.7 ± 0.1 ^b^	-	0.8 ± 0.0 ^c^	-	down
*TLR5*	1.0 ± 0.0 ^b^	4.1 ± 0.4 ^a^	up	1.1 ± 0.1 ^b^	-	0.8 ± 0.1 ^b^	-	down	0.7 ± 0.1 ^b^	-	0.7 ± 0.1 ^b^	-	down
*TLR6*	1.0 ± 0.0 ^b^	2.5 ± 0.4 ^a^	up	1.0 ± 0.1 ^b^	-	0.7 ± 0.1 ^b^	-	down	0.7 ± 0.1 ^b^	-	1.3 ± 0.3 ^b^	-	-
*TLR7*	1.0 ± 0.0 ^a^	1.5 ± 0.1 ^a^	-	0.9 ± 0.1 ^a^	-	0.7 ± 0.1 ^a^	-	-	0.7 ± 0.1 ^a^	-	1.2 ± 0.1 ^b^	-	-
*TLR8*	1.0 ± 0.0 ^b^	1.8 ± 0.1 ^a^	up	1.1 ± 0.1 ^b^	-	0.8 ± 0.1 ^b^	-	down	0.6 ± 0.1 ^b^	down	1.1 ± 0.1 ^b^	-	down
*TLR9*	1.0 ± 0.0 ^b^	1.6 ± 0.1 ^a^	up	1.0 ± 0.1 ^b^	-	1.2 ± 0.1 ^b^	-	down	1.0 ± 0.1 ^b^	-	1.5 ± 0.1 ^a^	up	-
*TLR10*	1.0 ± 0.0 ^a^	1.0 ± 0.2 ^a^	-	1.0 ± 0.2 ^a^	-	1.1 ± 0.2 ^a^	-	-	0.9 ± 0.1 ^a^	-	0.8 ± 0.1 ^a^	-	-

* Cocultured Caco2/HT-29MTX cells were pretreated with ME and ZnO for 2 h and challenged with LPS, as described in Section 4 (Materials and Methods). At the end of the experiment, the cells were collected and the gene expression was quantified by qPCR array. ** The results are presented as the mean ± SEM. ^#^ The effects of the experimental treatments on gene expression were analyzed by one-way ANOVA followed by Fisher’s tests, with *p* < 0.05 considered as statistically significant. ^a,b,c^ Groups with different superscript letters are significantly different (*p* < 0.05), where “a” = the highest value. up = upregulation; down = downregulation; - = no effect; “a” = The highest value.

**Table 4 ijms-26-00273-t004:** The effects of ME and ZnO on the signaling mediator’s gene expression in LPS-challenged Caco2/HT-29 cocultured cells.

Genes	Experimental Treatment *												
	Control	LPS		ME		ME+LPS			ZnO		ZnO+LPS		
	Mean ± SEM **	Mean ± SEM **	Compared to Control ^#^	Mean ± SEM **	Compared to Control ^#^	Mean ± SEM **	Compared to Control ^#^	Compared to LPS ^#^	Mean ± SEM **	Compared to Control ^#^	Mean ± SEM **	Compared to Control ^#^	Compared to LPS ^#^
*MyD88*	1.0 ± 0.0 ^b^	4.2 ± 0.4 ^a^	up	0.8 ± 0.0 ^b^	-	0.4 ± 0.1 ^c^	down	down	0.9 ± 0.1 ^b^	-	1.0 ± 0.1 ^b^	-	down
*MD-2*	1.0 ± 0.0 ^b^	4.3 ± 0.3 ^a^	up	0.9 ± 0.1 ^b^	-	1.0 ± 0.2 ^b^	-	down	0.5 ± 0.1 ^c^	down	0.5 ± 0.1 ^c^	down	down
*TRAF6*	1.0 ± 0.0 ^b^	3.0 ± 0.2 ^a^	up	1.3 ± 0.3 ^a^	-	0.9 ± 0.1 ^b^	-	down	1.2 ± 0.1 ^b^	-	1.3 ± 0.3 ^a^	-	down
*IRAK1*	1.0 ± 0.0 ^b^	2.6 ± 0.1 ^a^	up	1.1 ± 0.2 ^b^	-	1.1 ± 0.1 ^b^	-	down	1.2 ± 0.2 ^b^	-	1.5 ± 0.2 ^c^	-	down
*NOD1*	1.0 ± 0.0 ^b^	2.4 ± 0.3 ^a^	up	1.1 ± 0.3 ^b^	-	1.4 ± 0.1 ^a b^	-	-	0.8 ± 0.1 ^b^	-	0.9 ± 0.1 ^b^	-	down
*Tollip*	1.0 ± 0.0 ^b^	2.3 ± 0.3 ^a^	up	0.8 ± 0.1 ^b^	-	0.6 ± 0.1 ^c^	down	down	0.8 ± 0.0 ^b^	-	0.7 ± 0.1 ^a^	down	down

* Cocultured Caco2/HT-29MTX cells were pretreated with ME and ZnO for 2 h and challenged with LPS, as described in Section 4 (Materials and Methods). At the end of the experiment, cells were collected and the gene expression was quantified by qPCR array. ** The results are presented as the mean ± SEM. ^#^ The effects of the experimental treatments on gene expression were analyzed by one-way ANOVA followed by Fisher’s tests, with *p* < 0.05 being considered statistically significant. ^a,b,c^ Groups with different superscript letters are significantly different (*p* < 0.05), where “a” is the highest value. up = upregulation; down = downregulation; - = no effect; “a” = the highest value.

**Table 5 ijms-26-00273-t005:** Mitochondrial potential after exposure of cell mitochondria to LPS and the ME and ZnO experimental treatments.

	Control	ME	ZnO	LPS	ME+LPS	ZnO+LPS	SEM
Live (%)	94.15 a	94.18 a	94.00 a	90.41 b	93.13 a	94.39 a	0.615
Depolarized/Live (%)	0.72 a	0.91 a	0.89 a	3.70 b	1.55 a	1.52 a	0.339
Depolarized/Dead (%)	0.21 a	0.24 a	0.20 a	1.39 b	0.27 a	0.47 a	0.191
Dead (%)	4.00 a	4.02 a	3.95 a	4.50 a	4.57 a	2.94 b	0.322

The values correspond to the mean ± SEM. Means without a common letter among the treatments are significantly different (*p* < 0.05), Student’s *t*-test.

**Table 6 ijms-26-00273-t006:** The effects of ME and ZnO on genes coding for signaling pathway molecules in LPS-challenged Caco2/HT-29 cocultured cells.

Gene	Experimental Treatment *												
	Control	LPS		ME		ME+LPS			ZnO		ZnO+LPS		
	Mean ± SEM **	Mean ± SEM **	Compared to Control ^#^	Mean ± SEM **	Compared to Control ^#^	Mean ± SEM **	Compared to Control ^#^	Compared to LPS ^#^	Mean ± SEM **	Compared to Control ^#^	Mean ± SEM **	Compared to Control ^#^	Compared to LPS ^#^
*NF-kB/p65*	1.0 ± 0.0 ^b^	3.6 ± 0.6 ^a^	up	0.9 ± 0.1 ^b^	-	0.9 ± 0.2 ^b^	-	down	0.7 ± 0.1 ^b^	-	0.9 ± 0.1 ^b^	-	down
*TAK1-MAPK*	1.0 ± 0.0 ^b^	1.8 ± 0.1 ^a^	up	0.7 ± 0.2 ^b^	-	0.9 ± 0.2 ^b^	-	down	0.9 ± 0.3 ^b^	-	0.7 ± 0.1 ^b^	-	down
*p-38 MAPK*	1.0 ± 0.0 ^b^	2.3 ± 0.2 ^a^	up	0.8 ± 0.2 ^b^	-	1.0 ± 0.0 ^b^	-	down	0.8 ± 0.3 ^b^	-	1.2 ± 0.1 ^b^	-	down
*PKC-A*	1.0 ± 0.0 ^b^	1.8 ± 0.1 ^a^	up	0.8 ± 0.2 ^b^	-	0.8 ± 0.2 ^b^	-	down	0.7 ± 0.2 ^b^	-	0.9 ± 0.3 ^b^	-	down
*PKC-D*	1.0 ± 0.0 ^b^	2.2 ± 0.3 ^a^	up	0.9 ± 0.1 ^b^	-	0.8 ± 0.2 ^b^	-	down	0.9 ± 0.1 ^b^	-	1.2 ± 0.1 ^b^	-	down

* Cocultured Caco2/HT-29MTX cells were pretreated with ME and ZnO for 2 h and challenged with LPS, as described in Section 4 (Materials and Methods). At the end of the experiment, the cells were collected and the gene expression quantified by qPCR array. ** The results are presented as the mean ± SEM. ^#^ The effects of the experimental treatments on the gene expression were analyzed by one-way ANOVA followed by Fisher’s tests, with *p* < 0.05 considered statistically significant. ^a,b^ Groups with different superscript letters are significantly different (*p* < 0.05), where “a” = the highest value. up = upregulation; down = downregulation; - = no effect.

## Data Availability

The data are the property of INCDBNA-IBNA Balotesti and are available upon request to the authors.

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
