# Peer review of "Mustard Meal Extract as an Alternative to Zinc Oxide for Protecting the Intestinal Barrier Against E. coli-Lipopolysaccharide Damage"

_ijms, 2024, doi:10.3390/ijms26010273_

Round 1

Reviewer 1 Report

Comments and Suggestions for Authors

All comments, suggestions and questions are available throught the manuscript.

Author Response

List of itemized changes made in response to points raised by the reviewer 1

We thank very much to the reviewer for the pertinent observations.

Reviewer #1 Comments:

1)

-comment:

 Abstract: Please authors, make clear your objective and short your sentence.

Answer: the entire abstract has been rewritten in the new version of the manuscript

The present study aimed to investigate the ability of an aqueous extract derived from a mustard seed meal to counteract the effects produced by E. coli endotoxin lipopolysaccharide (LPS) on intestinal epithelium. Caco-2 cells were culture together with HT29-10 MTX and used as a cellular model to analysed important intestinal parameters such as renewal, integrity, innate immunity and signalling pathway. Mustard by-products resulted after the mustard oil extraction are rich in soluble polysaccharides, proteins, allyl isothiocyanates and phenolic acids, known as a powerful antioxidant with antimicrobial and antifungal properties. Cells were seeded in a ratio of 9 (Caco-2) to 1 (HT29-MXT) 17 and treated 2h with mustard meal extract (ME, dilution 1/50) and ZnO (50 μM) after reaching 80-18 100% of confluence. Then, cells were challenged with 5μg/mL E. coli-LPS and incubated for another 4h. The results showed that LPS did not altered cells viability, but decreased their proliferation when compared either to control or ME and ZnO treatments. LPS altered also the epithelial membrane integrity and consequently epithelium permeability by decreasing transepithelial electrical resistance and tight junction protein expression. On the other hand, LPS increased LDH activity and Toll like receptors expression. The mechanism by which LPS induces all these disturbances involved PKC, p38 MAPK, and NF-κB signalling molecules which were overexpressed. Pretreatment with mustard meal and ZnO succeeded in counteracting the epithelial renewal impairment, membrane integrity damage and decreased permeability. They also restore the gene expression of tight junction proteins.

2)

-comment:

What are your principal conclusions?

Answer: the revised abstract contains the study's conclusions.

What are the substances demonstrated negative effects to LPS?

Answer: Many studies in the literature have demonstrated that polyphenols and allyl isothiocyanates have antibacterial effects. The extract that we use was a polyphenolic extract.

3)

- comment:

Please authors, you can add to keywords: antioxidants, intestine epithelium, anti-inflammation, antibacterial.

Answer: the keywords: antioxidants, intestine epithelium, anti-inflammation, antibacterial were added in the new version of the manuscript.

4)

-comment:

Please authors, rewrite short sentence short maintaining your ideia. Review your sentences.

Answer: lines 59-63 have been shortened. The entire introduction has been rewritten for the new version of the manuscript.

5)

-comment:

Please authors, rewrite you sentence in affirmative form.

Answer: as suggested the sentence was rewrite in an affirmative form.

6)

-comment:

Please authots, cite them.

Answer: as suggested the bacterial strains were cited in the new version of the manuscript.

7)

-comment:

Please authors, uniforme your citation form according to author guide.

Answer: the references/citations were corrected according to author guide.

8)

-comment:

Too much long.

Answer: as suggested, the paragraph including lines 99-103 was rewritten.

9)

-comment:

Please authors, figures are small.

Answer: as suggested Figure 2 has been enlarged.

10)

-comment:

Please authors, figures are small.

Answer: as suggested Figure 3 has been enlarged.

11 and 12)

-comment:

Highlight text

Answer: as suggested texts were highlight.

13)

-comment:

Please authors, what these images are meaning?

14)

-comment:

Please authors, in what conditions polyphenols act against bacteria, inflammation and ROS?

  1. Polyphenols can act during bacterial invasion through different mechanism:

They can interact with the lipid components of bacterial cell membranes, disrupting their integrity, which leads to the loss of intracellular contents, compromising bacterial metabolism and bacterial death. Among polyphenols that possess antibacterial activity, there is Luteolin. Luteolin is considered a natural antibiotic because it can destroy the cell membrane of bacteria, inhibit nucleic acid synthesis, and modulate protein expression and energy metabolism.

Also, infectious diseases caused by microbial biofilms are a major clinical problem. It was shown that the new antimicrobial agents can inhibit biofilm formation and eradicate pre-formed bio-film.

Polyphenols actively inhibits enzymes involved in bacterial metabolic processes such as DNA replication, protein synthesis and cell wall formation. Ex. Inhibition of DNA gyrase or beta-lactamases produced by antibiotic-resistant bacteria.

  1. Polyphenols can act under stress conditions. Phytochemicals, abundant in plants, among them polyphenols, offer diverse antioxidant mechanisms. They scavenge free radicals, chelate metal ions, and modulate cellular signalling pathways to mitigate oxidative damage.

Also, polyphenols can act in hypoxia condition. For example, the resveratrol might act as a potent radiosensitizer and be a useful adjuvant agent against radiotherapy-resistant hypoxic cells in solid tumors.

  1. Many studies have shown that polyphenols act against inflammation by decreasing the synthesis of powerful pro-inflammatory cytokines such as TNF-a, IL-1beta, Il-6, IFN-g, etc.

Reviewer 2 Report

Comments and Suggestions for Authors

The overall concept and idea of the study are interesting, the method selection is mostly appropriate (with minor weaknesses). The study could be accepted for publication, but needs significant efforts to improve the quality:

1) The figures, specifically the bar graphs, are very poorly designed and lack essential elements like appropriate axis and labels. 

2) An issue I noticed in the manuscript is related to using comma for decimal place. 

3) The tables need to be formatted for better visual appearance.  

4) Section 2.1 - The units of the compounds are confusing with an inappropriate use of %. In addition, these values need to be compared with previous studies.

5) Authors need to be careful with the scientific forms of molecules. For example, line 380, H2O2 needs to be written as H2O2.

6) Section 4.1 - This needs further details especially with respect to analysis of phenolic compounds. 

7) A weakness of this study is that the mustard meal extract was directly applied to the cells. Realistically, in real life, the composition of the extract will change when it reaches the intestine. Thus, it would be better if the authors considered in vitro digestion of the extract and then applying it on cells and conducting the rest of the experiments.  

Comments on the Quality of English Language

I would rate it 3/5, punctuation needs a lot of care. 

Author Response

List of itemized changes made in response to points raised by the reviewer 2

We thank very much to the reviewer for the pertinent observations.

1)

-comment:

The figures, specifically the bar graphs, are very poorly designed and lack essential elements like appropriate axis and labels. 

Answer: As suggested all the graph were redesign.

2)

-comment:

An issue I noticed in the manuscript is related to using comma for decimal place. 

Answer: sorry for the mistake. The comma used for decimal was replace.

3)

-comment:

The tables need to be formatted for better visual appearance.  

Answer: As suggested all the tables were formatted

4)

-comment:

Section 2.1 - The units of the compounds are confusing with an inappropriate use of %. In addition, these values need to be compared with previous studies.

Answer: The concentration of each individual polyphenol was expressed as mg polyphenol compound/100g of mustard meal sample. However, to avoid any confusion, in the new version of the manuscript, the results are expressed in mg phenolic compound/g of mustard meal sample.

5)

-comment:

5) Authors need to be careful with the scientific forms of molecules. For example, line 380, H2O2 needs to be written as H2O2.

Answer: correction was done

6)

-comment:

Section 4.1 - This needs further details especially with respect to analysis of phenolic compounds. 

Answer: as suggested more detailed were provided for the section 4.1

7)

-comment:

A weakness of this study is that the mustard meal extract was directly applied to the cells. Realistically, in real life, the composition of the extract will change when it reaches the intestine. Thus, it would be better if the authors considered in vitro digestion of the extract and then applying it on cells and conducting the rest of the experiments.  

Answer: thank you for the reviewer’s suggestion.  This study was followed by an in vivo study on a porcine model in which the active components of mustard meal did not reach the cells directly, but were subjected to a real digestion as suggested by the reviewer. However, the results obtained confirmed the in vitro results. This valuable observation was added to the conclusions of the study.

-Comments on the Quality of English Language

I would rate it 3/5, punctuation needs a lot of care. 

The English has been reviewed by a native English speaker and in addition, authors will request an English check through the MDPI English service.

Reviewer 3 Report

Comments and Suggestions for Authors

Dear Authors, The results section of this manuscript reflects that the whole team has done a great job collecting the data, which is incredible. However, this manuscript needs rewriting from the beginning.

The introduction section is very scattered, the authors are unclear about what content to discuss in what sequence. Language is very very primitive and hard to understand.

The methodology is incomplete and only a few information are provided. Methodlgy should be written in such a way that new researcher can follow your method to reproduce the data.

I have provided several points. Please go through it and rewrite the manuscript by studying similar high-impact articles.

Abstract Section

Comment-1- Line 1-3- The article title looks so complex to understand. I think authors should rewrite it to simplify it. Also, it is not scientific to put an abbreviation in the title. My suggestion is “Mustard Meal Extract as an Alternative to Zinc Oxide for Protecting the Intestinal Barrier Against E. coli-lipopolysaccharide Damage”.

Comment-2-Line 18- treated 2 hours should be corrected as treated for 2 hours.

Comment-3- Line 19- give the full form of LPS for the first appearance and the same for Zno.

Comment-4- Line 22- “LPS altered also the epithelial membrane integrity “rewrite this.

Comment-5- “LPS altered also the epithelial membrane integrity and consequently epithelium 22 permeability by decreasing transepithelial electrical resistance and tight junction protein expression 23 and on the other hand by increasing LDH activity and Toll-like receptors expression. “Should rewrite this paragraph to make it more standard.

Comment-6- “negative effects” For scientific articles these types of generic terms do not sound great. What does negative effects mean, it can be specified particularly.

Introduction Section

Comment-7- Line 28, in the keyword section I suggest adding the term endotoxin lipopolysaccharide also.

Comment-8- Line -40- In the introduction section authors strongly suggested making a detailed discussion about endotoxin LPS, focusing on E. coli-derived LPS, emphasizing their chemical composition, possible toxicity, and mechanism.

Comment-9- Line 56- “Also, [8] reported the successful “This is not the way of writing. It can simply be written as “a previous study reported that “ as well as “Literature contains also data -line 65“Rewrite this as well.

Comment-10- Line -73- In the text of the manuscript we never put the whole website link (https://www.feedipedia.org). It should be in the reference list. Also, information from random websites is not trustworthy and can’t be considered an authentic reference. So, authors should search for similar kinds of published papers.

Comment-11- Line 78- “For example, [11] “ This is not the way of writing and citing the reference. First, complete the sentence and cite the reference at the end. Also, For example has been repeated many times in the manuscript. Overall, the way of writing is poor, it needs revision.

Comment-12- Line 76- “There are studies which prove that the active phytocompounds from mustard possess “Which part of mustard, is it oil, seed extract, or any other specific part?

Comment-13- Line 76-92- In this paragraph, if authors want to highlight the medicinal value of mustard xxxx, authors should highlight that all kinds of biological activities have been reported till now. However, authors are mainly emphasizing antibacterial and even writing the numerical values of ZOI. This kind of information might be suitable in the discussion section. The introduction is to give literature background on your topic not to discuss the results of previous papers. Rather in a short paragraph authors can justify their aim by saying that although several studies have been reported in other parts of mustard, the research in byproducts is scant. So, our research is aimed for --------.

Comment-14-Lin 94-104- In the previous paragraph, after claiming why they are doing this research authors are discussing Caco-2, which looks unusual.

Comment-15- Line 99-104- This information must already be in the methodology section. So, it is irrelevant to put it here.

Comment-16- Overall, the way of writing in the introduction does not look organized. The information in the introduction is so much scattered. Authors should make visible the flow of information rather than writing randomly. I strongly suggest rewriting the introduction again by reading similar published high-impact scientific papers,

Result Section

Comment-17- HPLC Cheromatorgram for polyphenol content should be included in the manuscript.

Comment 18-Line 109-110- “highest concentration was 109 registered for rutin (2119,96 mg%g)” What does this mean?  All the concentrations expressed in polyphenol content are wrong.

Comment-19- Line 114-115- “Epithelial renewal, one of the key characteristics of the intestinal epithelium was as- 114 assessed through cell viability, proliferation, cell cycle and apoptosis” This sentence should not be in the 2.1.

Comment-20- All the tables in the results are constructed very poorly. Lots of words are missing. Tabulation should be remade throughout the manuscript.

Comment-21- Figure 1 and Figure 5 should be remade with better quality.

Comment-22- Line 78-79- “filtered and injected 478 into HPLC to measure the polyphenols” How authors can determine the polyphenol content quantitatively if direct filtrate from the extraction was injected. How do authors know the amount of extract present in that filtrate? What was the injection volume? The whole methodology should be described clearly.

Comment-23- Line 481- “Aqueous extract was diluted, sterile filtered and 481 used in cell co-cultures”- What does this mean? Initially, authors said they extracted in methanol. How aqueous extract was mentioned suddenly.

Comments on the Quality of English Language

The quality of English is very very poor. Mandatorily it needs extensive English improvement service. It is really hard to improve this English by the same authors.

Author Response

List of itemized changes made in response to points raised by the reviewer 3

Abstract Section

Comment-1- Line 1-3- The article title looks so complex to understand. I think authors should rewrite it to simplify it. Also, it is not scientific to put an abbreviation in the title. My suggestion is “Mustard Meal Extract as an Alternative to Zinc Oxide for Protecting the Intestinal Barrier Against E. coli-lipopolysaccharide Damage”.

Answer:  As suggested the title was change accordingly.

Comment-2-Line 18- treated 2 hours should be corrected as treated for 2 hours.

Answer: correction was done.

Comment-3- Line 19- give the full form of LPS for the first appearance and the same for Zno.

Answer: the full form of LPS is given above, line 12.

Comment-4- Line 22- “LPS altered also the epithelial membrane integrity “rewrite this.

Answer: the correction was done.

Comment-5- “LPS altered also the epithelial membrane integrity and consequently epithelium 22 permeability by decreasing transepithelial electrical resistance and tight junction protein expression 23 and on the other hand by increasing LDH activity and Toll-like receptors expression. “Should rewrite this paragraph to make it more standard.

Answer: the correction was done. The entire abstract was rewrite.

Comment-6- “negative effects” For scientific articles these types of generic terms do not sound great. What does negative effects mean, it can be specified particularly.

Answer: the correction was done. The entire abstract was rewrite.

Introduction Section

Comment-7- Line 28, in the keyword section I suggest adding the term endotoxin lipopolysaccharide also.

Answer: As suggested the term endotoxin lipopolysaccharide was added among the keyword.

Comment-8- Line -40- In the introduction section authors strongly suggested making a detailed discussion about endotoxin LPS, focusing on E. coli-derived LPS, emphasizing their chemical composition, possible toxicity, and mechanism.

Answer: As suggested a paragraph regarding LPS was added in the new version of the manuscript.

Comment-9- Line 56- “Also, [8] reported the successful “This is not the way of writing. It can simply be written as “a previous study reported that “ as well as “Literature contains also data -line 65“Rewrite this as well.

Answer: as suggested the sentences were rewrite.

Comment-10- Line -73- In the text of the manuscript we never put the whole website link (https://www.feedipedia.org). It should be in the reference list. Also, information from random websites is not trustworthy and can’t be considered an authentic reference. So, authors should search for similar kinds of published papers.

Answer: the correction was done. A similar reference was added.

Comment-11- Line 78- “For example, [11] “ This is not the way of writing and citing the reference. First, complete the sentence and cite the reference at the end. Also, For example has been repeated many times in the manuscript. Overall, the way of writing is poor, it needs revision.

Answer: the correction was done. Expression “For example” 47 / 5.000

The expression “for example” was avoided in use. The English has been revised by a native English speaker and in addition, authors will request an English check through the MDPI English service.

Comment-12- Line 76- “There are studies which prove that the active phytocompounds from mustard possess “Which part of mustard, is it oil, seed extract, or any other specific part?

Answer: Details about details regarding the mustard source that was used in the mentioned studies have been added to the paragraph. 

Comment-13- Line 76-92- In this paragraph, if authors want to highlight the medicinal value of mustard xxxx, authors should highlight that all kinds of biological activities have been reported till now. However, authors are mainly emphasizing antibacterial and even writing the numerical values of ZOI. This kind of information might be suitable in the discussion section. The introduction is to give literature background on your topic not to discuss the results of previous papers. Rather in a short paragraph authors can justify their aim by saying that although several studies have been reported in other parts of mustard, the research in byproducts is scant. So, our research is aimed for --------.

Answer: As suggested The paragraph was shortened, but certain details were kept because the other two reviewers requested that the names of the bacteria on which the mustard phytocompounds have an effect be specified.

Comment-14-Lin 94-104- In the previous paragraph, after claiming why they are doing this research authors are discussing Caco-2, which looks unusual.

Answer: the paragraph was eliminated.

Comment-15- Line 99-104- This information must already be in the methodology section. So, it is irrelevant to put it here.

Answer: As suggested the information from this paragraph was moved in the methodology section.

Comment-16- Overall, the way of writing in the introduction does not look organized. The information in the introduction is so much scattered. Authors should make visible the flow of information rather than writing randomly. I strongly suggest rewriting the introduction again by reading similar published high-impact scientific papers,

We did our best to improve the introduction which was rewritten and the English has been reviewed by a native English speaker and in addition, authors will request an English check through the MDPI English service.

Result Section

Comment-17- HPLC Cheromatorgram for polyphenol content should be included in the manuscript.

Answer: HPLC Chromatogram for polyphenols was added in the new version of the manuscripts.

Comment 18-Line 109-110- “highest concentration was registered for rutin (2119,96 mg%g)” What does this mean?  All the concentrations expressed in polyphenol content are wrong.

Answer: The concentration of each individual polyphenol was expressed as mg polyphenol compound/100g of mustard meal sample. However, to avoid any confusion, in the new version of the manuscript, the results are expressed in mg phenolic compound/g of mustard meal sample.

Comment-19- Line 114-115- “Epithelial renewal, one of the key characteristics of the intestinal epithelium was assessed through cell viability, proliferation, cell cycle and apoptosis” This sentence should not be in the 2.1.

Answer: Thank you for the valuable observation. The sentence was moved from 2.1.

Comment-20- All the tables in the results are constructed very poorly. Lots of words are missing. Tabulation should be remade throughout the manuscript.

Answer: Thank you for the valuable observation. all tables have been replaced with editable form

Comment-21- Figure 1 and Figure 5 should be remade with better quality.

Answer: Thank you for the valuable observation. All the figures

Comment-22- Line 78-79- “filtered and injected into HPLC to measure the polyphenols” How authors can determine the polyphenol content quantitatively if direct filtrate from the extraction was injected. How do authors know the amount of extract present in that filtrate? What was the injection volume? The whole methodology should be described clearly.

Answer: the method for phenolic compounds measurement was entirely revised. Also the whole methodology involved in this study was revised.

Comment-23- Line 481- “Aqueous extract was diluted, sterile filtered and 481 used in cell co-cultures”- What does this mean? Initially, authors said they extracted in methanol. How aqueous extract was mentioned suddenly.

 Answer: the correction was done in the new version of material and methods.

Five grams of mustard seed meal was mixed with 35 ml of extraction solvent (methanol 80%) ratio of sample:solvent (1:7 w/v) and shaken continuously overnight. The supernatant was recovered by centrifugation at 4500 rpm for 20 min and filtered. To be used in cellular culture, the methanolic supernatant was concentrated with a rotary vacuum concentrator (CHRIST AVC 2-18CO Plus, Germany) to remove the methanol and 1 ml of water was added to the remaining organic residue. Aqueous extract was sterile filtered, diluted and used in cell co-cultures. ZnO solution was prepared as described by [13].

 Also, the whole methodology involved in this study was revised.

Comments on the Quality of English Language

The quality of English is very very poor. Mandatorily it needs extensive English improvement service. It is really hard to improve this English by the same authors.

Answer: authors will request an English check through the MDPI English service.

Round 2

Reviewer 2 Report

Comments and Suggestions for Authors

The authors have addressed all the comments, except for the comment related to the quality of the bar graphs. There are still issues with the quality of the bar graphs including the title, using commas instead of decimal places, and punctuation issues within the figure.

Author Response

Comments and Suggestions for Authors

The authors have addressed all the comments, except for the comment related to the quality of the bar graphs. There are still issues with the quality of the bar graphs including the title, using commas instead of decimal places, and punctuation issues within the figure.

Answer: All bar graphs have been revised and no longer contain commas. All graph titles have also been standardized.

Reviewer 3 Report

Comments and Suggestions for Authors

Comment-1-Line 540-555- In result 2.1, authors have expressed their results as mg/g of dried extract. That means the authors should have obtained dried extract before injecting the sample. So, the authors should rewrite and indicate they dissolved dried extract in water and also mention that what was the concentration of extract solution injected in HPLC.

Comment-2. based on the HPLC Chromatogram of 2.1 in the result section, it can't be assured those peaks are of corresponding compounds. The resolution of the peak is inferior and that peak may contain other impurities also. Either the author should have to put a better chromatogram or the authors should remove this section, as this level of chromatogram doesn't suit this level of the journal.

3. The authors have mentioned that the manuscript will go MDPI English editing service, which I strongly recommend.

2GAE/4.1

Obtention of mustard meal extract and ZnO solution

4 IN.1

Obtention of mustard meal extract and ZnO solution

Comments on the Quality of English Language

The manuscript should undergo the English editing service of MDPI

Author Response

Comments and Suggestions for Authors

Comment-1-Line 540-555- In result 2.1, authors have expressed their results as mg/g of dried extract. That means the authors should have obtained dried extract before injecting the sample. So, the authors should rewrite and indicate they dissolved dried extract in water and also mention that what was the concentration of extract solution injected in HPLC.

Comment-2. based on the HPLC Chromatogram of 2.1 in the result section, it can't be assured those peaks are of corresponding compounds. The resolution of the peak is inferior and that peak may contain other impurities also. Either the author should have to put a better chromatogram or the authors should remove this section, as this level of chromatogram doesn't suit this level of the journal.

Answer for comment 1+2: as recommended to avoid all errors and misunderstandings, figure 1 (HPLC chromatogram) was removed. Only the concentration of total polyphenols determined by the Folin-Ciocalteu method was kept because it quantifies all polyphenols including all peaks unidentified by us. Antioxidant activity measured by using ABTS assay was also maintained.

  1. The authors have mentioned that the manuscript will go MDPI English editing service, which I strongly recommend.

Answer: English proofreading via MDPI service has been enabled.